# An evaluation of the E3SMv1-Arctic Ocean/Sea Ice Regionally Refined Model

Milena Veneziani[1], Wieslaw Maslowski[2], Younjoo, J. Lee[2], Gennaro D'Angelo[1], Robert Osinski[3], Mark, R. Petersen[1], Wilbert Weijer[1], Anthony, P. Craig[4], John, D. Wolfe[1], Darin Comeau[1], and Adrian, K. Turner[1]

[1]Los Alamos National Laboratory, Los Alamos, NM, USA
[2]Naval Postgraduate School, Monterey, CA, USA
[3]Institute of Oceanology of Polish Academy of Sciences, Sopot, Poland
[4]independent researcher

**Correspondence:** Milena Veneziani (milena@lanl.gov)

**Abstract.** The Energy Exascale Earth System Model (E3SM) is a state-of-the-science Earth system model (ESM) with the ability to focus horizontal resolution of its multiple components in specific areas. Regionally refined global ESMs are motivated by the need to explicitly resolve, rather than parameterize, relevant physics within the regions of refined resolution, while offering significant computational cost savings relative to the respective cost of configurations with high-resolution (HR) everywhere on the globe. In this paper, we document results from the first Arctic regionally refined E3SM configuration for the ocean and sea-ice components (E3SM-Arctic-OSI), while employing data-based atmosphere, land, and hydrology components. Our aim is an improved representation of the Arctic coupled ocean and sea ice state, its variability and trends, and the exchanges of mass and property fluxes between the Arctic and the Subarctic. We find that E3SM-Arctic-OSI increases the realism of simulated Arctic ocean and sea ice conditions compared to a similar low-resolution E3SM simulation without the Arctic regional refinement in ocean and sea ice components (E3SM-LR-OSI). In particular, exchanges through the main Arctic gateways are greatly improved with respect to E3SM-LR-OSI. Other aspects, such as the Arctic freshwater content variability and sea-ice trends, are also satisfactorily simulated. Yet, other features, such as the upper ocean stratification and the sea-ice thickness distribution, need further improvements, involving either more advanced parameterizations, model tuning, or additional grid refinements. Overall, E3SM-Arctic-OSI offers an improved representation of the Arctic system relative to E3SM-LR-OSI, at a fraction (15%) of the computational cost of comparable global high-resolution configurations, while permitting exchanges with the lower latitude oceans that can not be directly accounted for in Arctic regional models.

## 1 Introduction

The Arctic Ocean has been undergoing fundamental changes over the past several decades, which are best exemplified by a drastic year-round, and particularly summer, decline in sea-ice coverage (Perovich et al., 2019). Given that sea-ice modulates the energy and property exchanges between the ocean and atmosphere, the observed decline of sea ice cover has impacted these interaction processes, their regional states, coupling and associated variability. Some of the key impacts of the sea ice decline

include an accumulation of heat absorbed in the upper Arctic Ocean (e.g., Timmermans et al., 2018), due to reduced surface albedo and a related amplified warming of the lower atmosphere (e.g., Dai et al., 2019) relative to the globally averaged rate of warming in response to increasing $CO_2$. In addition, several studies have ascribed the anomalous persistence of the anticyclonic

Beaufort Gyre since 1997 until present day – and the resulting continuous accumulation of freshwater within the Beaufort Gyre region – to the decline of sea ice (Proshutinsky et al., 2009; Rabe et al., 2014; Haine et al., 2015; Proshutinsky et al., 2019). The freshwater accumulation in the Arctic Ocean and its export through the Canadian Archipelago and Fram Strait is of relevance to the global ocean thermohaline circulation because of its potential impact on convection and deep water formation in the Greenland, Iceland, Irminger and Labrador seas (Häkkinen, 1993; Zhang et al., 2021).

Furthermore, the Arctic sea ice and climate are influenced by northward advection of warm water from the Pacific and Atlantic oceans. Polyakov et al. (2017) have recently introduced the concept of 'Atlantification' of the Arctic, recognizing an increasing impact of incoming Atlantic waters entering the eastern basin through the Barents Sea Opening (BSO) and Fram Strait on the sea ice cover and the upper ocean stratification downstream, which acts to increase winter ventilation in the ocean interior. Similarly, on the Pacific side, A. Woodgate and Peralta-Ferriz (2021) have reported an increasing inflow

and warming of waters transported northward across Bering Strait during 1990-2019, which amplifies their impact on the ice regime downstream in the western Arctic Ocean, where the ice has retreated furthest north in recent summers.

The above examples and many other Arctic to mid-latitude exchange processes are inherently associated with feedbacks between various components of the Earth System, namely the ocean, cryosphere, atmosphere, and land hydrology, and are therefore better explored using a global, fully coupled Earth System Model (ESM). One such model is the recently developed

Energy Exascale Earth System Model (E3SM), sponsored by the United States Department of Energy (Golaz et al., 2019). To our knowledge, there is only one other Arctic regionally refined ESM configuration to date, i.e. the Finite Element Sea ice-Ocean Model (FESOM; Wekerle et al., 2013, 2017a, b; Wang et al., 2018). The ocean and sea-ice model components of E3SM are based on the unstructured-grid Model for Prediction Across Scales (MPAS) framework; hence they are particularly suited for focusing resolution in specific regions toward explicitly resolving fine-scale physics, rather than parameterizing it, while

retaining the context of a global Earth System configuration (Ringler et al., 2013; Petersen et al., 2019). In this study, we utilize the E3SM-MPAS framework to evaluate the first regionally refined E3SM Arctic ocean/sea-ice configuration with a data-based atmosphere model component (E3SM-Arctic-OSI), using 10 km horizontal resolution in the pan-Arctic region and $10-60$ km resolution elsewhere. A similar configuration to this, but with Arctic regional refinement of 6 km was also considered initially; that simulation, while being approximately three times more expensive than the one described in this paper, did not produce

any significant improvements in the Arctic and subarctic ocean and sea-ice representation. We concluded that a resolution of at least 3 km is necessary to really resolve the local Rossby radius of deformation in most of the Arctic, and we plan to actively work on such very high resolution E3SM-Arctic configuration in the near future. While more specific studies using this model will follow, we deem important to document this first effort towards Arctic regional refinement in E3SM, which ultimately will include comparable grid refinements in the atmosphere and land components of E3SM.

The main objective of the paper is to investigate whether enhanced resolution in the Arctic and Subarctic translate into an improved simulation of the sea-ice cover, the oceanic conditions, and the Arctic-Subarctic exchanges through the main Arctic

gateways. For the reasons mentioned previously in this section, we are interested not only in the pan-Arctic but also in the simulation of global and large scale metrics such as the Atlantic Meridional Overturning Circulation (AMOC). We achieve this main objective by comparing E3SM-Arctic-OSI with a companion forced E3SM ice-ocean simulation that uses a global low-resolution mesh (E3SM-LR-OSI). We also compare results with a high-resolution Regional Arctic System Model (RASM) simulation when observations are scarce or unavailable. Due to the higher number of constraints and its Arctic focus, we expect RASM to give a realistic representation of local processes, while obviously not directly accounting for the Arctic to mid-latitude exchange processes. This study is expected to provide important insights to future model configurations by the E3SM and by the broader Arctic modeling community. A secondary objective of the present paper is to document Arctic-focused model evaluation metrics for E3SM. This is accomplished through both common scripts for standalone model-observation comparisons and by the addition of Arctic metrics to the MPAS-Analysis package, which is a python-based analysis package developed at the Los Alamos National Laboratory specifically for MPAS model components[1].

The paper is organized as follows: a description of the model configurations and simulations utilized throughout the paper is included in Section 2. Results in terms of both global diagnostics and Arctic focused metrics are presented in Sections 3-5. Finally, a discussion and conclusions are included in Section 6.

## 2 Model Configurations and Simulations

In this section, we provide some details of the following three model configurations: E3SM-Arctic-OSI, E3SM-LR-OSI, and the RASM simulation.

The ocean and sea-ice model components of E3SM are MPAS-Ocean and MPAS-Seaice (Petersen et al., 2019; Turner et al., 2022), respectively, and the two components share a common mesh that is typically made by hexagonal grid elements, although cells may have any number of sides. The MPAS-Ocean vertical grid is structured and consists of 80 levels, with vertical resolution ranging between 2 m in the upper 10 m of the water column and 200 m towards the ocean bottom. The configurations presented here use z-star, where the layer thicknesses of the full column expand and contract with the sea surface height (Adcroft and Campin, 2004; Petersen et al., 2015). The ocean prognostic volume equation of state includes surface fluxes from the atmosphere and land via the coupler; thus, virtual salinity fluxes are not utilized.

The E3SM-Arctic-OSI configuration has horizontal resolution of 10 km in the Arctic Ocean, whereas, in the southern hemisphere, it has the nominal horizontal resolution of $1°$ that is the E3SM standard low-resolution used in E3SM-LR-OSI and in the model study of Golaz et al. (2019) (Fig. 1b). As seen in Fig. 1a (red curves), the mesh resolution transitions from 60 km in the Southern Hemisphere to 30 km in the tropics to 10 km north of $60°$ N (for this reason, the E3SM-Arctic-OSI mesh is also referred to as 60to10). The horizontal resolution transitions to smaller grid cells more quickly in the Atlantic than in the Pacific Ocean (compare solid versus dashed lines in Fig. 1a), ensuring that (i) the Gulf Stream extension region (around $40°$ N) is characterized by a resolution of at least 15 km, and (ii) that the subpolar North Atlantic (north of $50°$ N) has a resolution similar to the one in the Arctic (around 10-12 km). Fig. 1c shows a zoom-in of the E3SM-Arctic-OSI mesh over the Arctic

---

[1]https://mpas-dev.github.io/MPAS-Analysis/stable/

and subpolar North Atlantic region, with a further enlargement inset displaying the hexagonal cells in more detail. The total number of cells is 0.62 M and the computational cost is 1.65 M cpu hours per simulated century. In comparison, the global high resolution E3SM configuration (E3SM-HR; Caldwell et al., 2019) has a computational cost of 11.17 M cpu hours per simulated century (Petersen et al., 2019); therefore, the E3SM-Arctic-OSI computational cost is about 15% the computational cost of E3SM-HR.

The ocean baroclinic time step is equal to 10 minutes. Ocean vertical mixing is parameterized through the K-profile parameterization method (KPP; Large et al., 1994), and no background vertical diffusivity is utilized. Mesoscale eddy effects are represented using the Gent-McWilliams (GM) eddy transport parameterization of Gent and McWilliams (1990) in regions outside of the Arctic and pan-Arctic. To achieve this regionally varying application of GM, a simple algorithm has been implemented in MPAS-Ocean for which the GM parameter is a ramp-like function of grid cell size. In particular for the E3SM-Arctic-OSI configuration considered here, the GM kappa parameter varies linearly between zero for cell sizes below 20 km and a maximum value of 600 $m^2\ s^{-1}$ for cell sizes above 30 km. This means that we effectively transition from GM-on to GM-off in the North Atlantic within $\approx 10° - 28°$ N, and approximately within $25° - 50°$ N elsewhere (see areas between the white and red lines in Fig. 1b). In other words, the GM kappa is 0 for latitudes above the red line and is equal to its maximum 600 $m^2\ s^{-1}$ for latitudes below the white line. Other parameterizations used in MPAS-Ocean are invariable with horizontal resolution.

The version of MPAS-Seaice used in this paper and the way that the sea ice and ocean components are coupled together are fully described in Turner et al. (2022) and Petersen et al. (2019); we have not changed any default MPAS-Seaice parameter for the purposes of the present effort.

The atmospheric data used to force the ocean and sea-ice model components is the Japanese atmospheric reanalysis product for driving ocean-sea-ice models (JRA55-do, version v1.3; Tsujino et al., 2018). At the time of our simulations, the JRA55-do atmosphere fluxes and river runoff data set was available for the period 1958-2016. The JRA55 product has a temporal resolution of 3-hours and a horizontal resolution of $0.5625°$, which is more than three times higher than the resolution of the Coordinated Ocean-ice Reference Experiment (CORE; Griffies et al., 2009) data used in forced ice-ocean ESM simulations until recently. Following Griffies et al. (2009), sea surface salinity (SSS) is restored to monthly climatological values obtained from the Polar science center Hydrographic Climatology (PHC3.0; updated from Steele et al., 2001), with an equivalent restoring time scale of 1 year.

The E3SM-LR-OSI configuration is similar to E3SM-Arctic-OSI in terms of atmospheric forcing, but it uses the standard global low-resolution mesh (black curve in Fig. 1a) and 60 vertical levels, with vertical resolution ranging between 10 m in the upper 200 m of the water column and 250 m below 3000 m depth. In this case, the GM parameterization is on at all latitudes. The ocean baroclinic time step is equal to 30 minutes. Key features of the E3SM configurations described above (and of the RASM simulation) have been summarized and compared against those described in three previous E3SM publications in Table 1.

We have performed two simulations consisting of three consecutive JRA cycles, one using E3SM-Arctic-OSI and one using E3SM-LR-OSI. The choice of three cycles was mostly constrained by the availability of computational resources when these simulations were performed. We also compared trends of fields of interest during the second and third cycles, and, as the

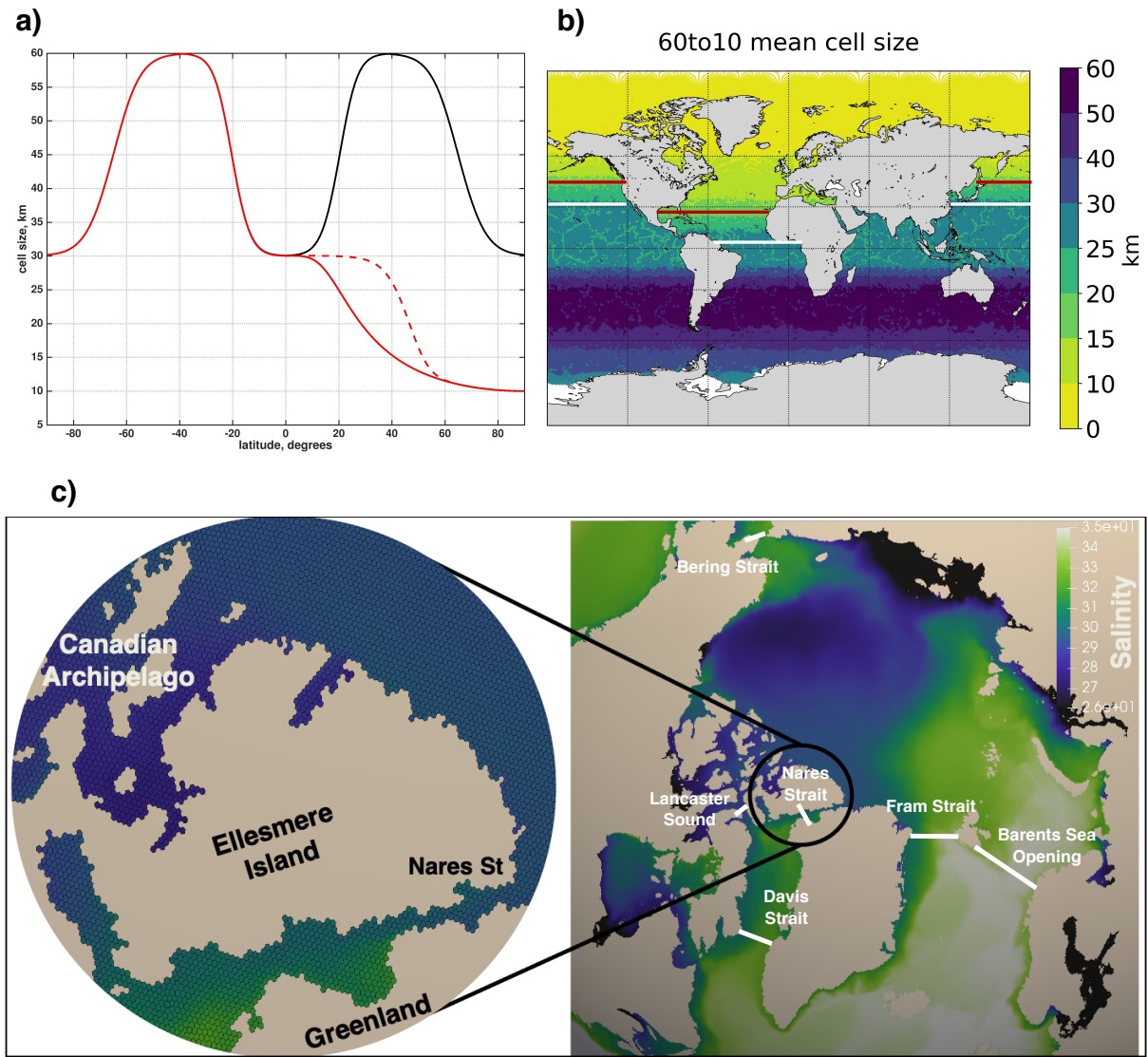

**Figure 1.** (a) Distribution function used to define cell size as a function of latitude and create the meshes: red curves indicate the E3SM-Arctic mesh, whereas the black curve indicate the E3SM-LR mesh. The solid red line marks resolution changes in the Atlantic Ocean and the dashed red line marks changes in the Pacific Ocean. Note that all lines converge (same behavior everywhere for both E3SM-Arctic and E3SM-LR) in the southern hemisphere. (b) Geographical distribution of grid cell size for the E3SM-Arctic-OSI configuration. The area between the white and red lines denotes the region where the transition between GM-on and GM-off occurs (no GM eddy parameterization is used north of the red lines). (c) Zoom-in around the Arctic of a salinity field simulated in E3SM-Arctic-OSI, with a further enlargement around Ellesmere Island to show the hexagonal mesh in more detail. Also shown are the locations of the five Arctic gateways and Davis Strait, through which fluxes in and out of the Arctic are later calculated.

**Table 1.** Main configuration differences between the model experiments described in this paper and those described in key E3SMv1 (version 1) publications. Numbers used in the 'Horizontal mesh' column refer to the minimum and maximum resolution in km for E3SM cases, while we have indicated that a regular 9 km resolution mesh is used for the RASM case.

| Study | Model | Atmosphere forcing | Horizontal mesh | Vertical levels | GM |
|---|---|---|---|---|---|
| This paper | E3SM-Arctic-OSI | JRA55-do | Arctic 60to10 | 80 | on outside Arctic |
| | E3SM-LR-OSI | JRA55-do | 60to30 | 60 | fully on |
| | RASM | JRA55-do | Regular 9km | 45 | fully off |
| Petersen et al. (2019) | E3SMv1-HR-OSI | CORE-II | 18to6 | 80 | fully off |
| | E3SMv1-LR-OSI | CORE-II | 60to30 | 60 | fully on |
| Golaz et al. (2019) | E3SMv1-LR | coupled | 60to30 | 60 | fully on |
| Caldwell et al. (2019) | E3SMv1-HR | coupled | 18to6 | 80 | fully off |

results shown later in the paper will elucidate, we were sufficiently satisfied that such trends remained mostly stable between the second and third cycle. In both simulations, the ocean is initialized from a one month spin up from rest, to allow for initial gravity waves adjustment, and from a temperature and salinity initial condition obtained from the PHC January climatology. Sea ice is initialized with a 1 m thick disc of sea ice extending to $60°$ N and S.

RASM is a fully-coupled, limited-area ESM, which has been used for dynamic downscaling of global atmospheric reanalyses as well as ESM projections (Maslowski et al., 2012; Roberts et al., 2015; Hamman et al., 2016; Cassano et al., 2017; Brunke et al., 2018). It includes the Weather Research and Forecasting (WRF) atmosphere model, the Parallel Ocean Program (POP) ocean component, the Community Ice Model (CICE) sea-ice component, and the Variable Infiltration Capacity (VIC) land hydrology model. A source-to-sink river routing model (RVIC) allows coupling of the land hydrology and ocean components. All these component models are coupled every 20 minutes using a version of the CESM coupler, CPL7, modified for a regional application. The model domain covers the entire pan-Arctic region (extending down to $\approx 45°$ N in the North Atlantic and to $\approx 30°$ N in the North Pacific), including the entire marine cryosphere of the Northern Hemisphere as well as terrestrial drainage to the Arctic Ocean and its margins. The RASM configuration used for the model intercomparison in sections 4 and 5 has a horizontal resolution of 9 km (i.e., $1/12°$ in a rotated spherical coordinate system) throughout the domain with 45 vertical levels. The sea ice component shares the same horizontal resolution as the ocean and it is configured with five ice thickness categories. The RASM ocean temperature and salinity along the closed later boundaries are restored to the monthly PHC3.0 climatology. No lateral boundary conditions for sea ice are required given the extent of the pan-Arctic domain. The RASM results used in this paper are from an ocean-sea ice simulation forced with JRA55-do, which in turn was initialized from a 75-year long spin up forced with the Coordinated Ocean-ice Reference Experiments Corrected Inter-Annual Forcing Version 2.0 (CORE2-CIAF); here, we focus on the last 40 years (1979-2018) of this run.

## 3 Global Ocean

The purpose of this section is to describe how the E3SM-Arctic-OSI simulation represents climatologies and trends of key global ocean fields. The global trend of OHC anomaly for three depth ranges and of T and S as a function of depth, are presented in Figure 2. Anomalies are computed relative to the first-year annual means and are 1-year running averaged to filter out the seasonal cycle. At the end of the third JRA cycle, the T and S distribution is only slightly trending in the 500-1000 m depth range. Alternating bands of warming and cooling are found in the upper 2000 m of the global water column (Fig. 2b), although the OHC for the $0 - 700$ m depth range indicates a net warming for the upper ocean (Fig. 2a). On the other hand, the bottom waters deeper than 4000 m exhibit a cooling persistent anomaly of up to $0.5°$C. Salinity experiences a more regular change with depth, with a freshening of up to $0.2$ psu in the upper 800 m and a salinification of up to $0.1$ psu in the deeper ocean (Fig. 2c). Overall, the top-to-bottom trends are all reduced during the third cycle, whereas the upper ocean warming and freshening are both still present towards the end of the simulation.

When comparing the model global T and S with observations (from the Roemmich-Gilson Argo data; Roemmich and Gilson, 2009) at different depths (surface, 150, 400, and 1500 m) in Figures 3-4, we mostly note the generally fresh bias in the upper 150 m, especially evident south of $40°$ S and in the North Atlantic at the surface, but also in the tropical Pacific at 150 m. This behavior is consistent with biases documented in Golaz et al. (2019). Recent improvements in the MPAS-Ocean eddy parameterization scheme have lead to a drastic reduction of the biases in the upper 1000 m ocean stratification, mainly in the Southern Ocean and Labrador Sea (not shown, unpublished results). T and S biases are much reduced below 1000 m (Fig. 4e,f).

Another important indication of the state of a global ESM is the AMOC. To that effect, the E3SM-Arctic-OSI overturning streamfunction plot as a function of latitude as well as the time series of maximum AMOC at $26.5°$ N for both the E3SM-Arctic-OSI and E3SM-LR-OSI, are shown in Figure 5. Observational variability from the Rapid Climate Change-Meridional Overturning Circulation and Heat-flux Array (RAPID-MOCHA) data set (Cunningham et al., 2007; McCarthy et al., 2020) is shaded in green for reference. While we have a reasonable (albeit on the strong side) Antarctic Bottom Water cell of $2 - 4$ Sv (e.g., Orsi et al., 2002), the upper cell of the AMOC at $26.5°$ N is weaker than observations by $\approx 6$ Sv in E3SM-Arctic-OSI, and is reduced by an additional 3 Sv in E3SM-LR-OSI (the average value from RAPID over the period 2004-2017 is $16.8 \pm 4.4$ Sv; McCarthy et al., 2020). A weak AMOC in other low-resolution E3SM simulations has been reported previously (Golaz et al., 2019; Weijer et al., 2020). Although a thorough investigation of its causes is beyond the purposes of this paper, we hypothesize that improved SSS and ocean stratification in the subpolar North Atlantic and in the Southern Ocean has an important impact on E3SM deep high-latitude convection and on its AMOC. SSS biases calculated similarly to Fig. 3 but for the E3SM-LR-OSI (not shown) are more than 1 psu fresher than those for E3SM-Arctic-OSI in the Nordic Seas region, and that is associated with higher mixed layer depth biases (not shown) with respect to an Argo-floats derived observational product. Such differences in stratification between E3SM-Arctic-OSI and E3SM-LR-OSI are clearly presented in Fig. 6, which shows a meridional vertical section of zonally averaged salinity in the Atlantic Ocean (north of Fram Strait the average is computed over the whole Arctic) with overlapped contours of sigma2 (potential density with respect to 2000 m). The fields are interannual averages

computed over years 148-177. Two main features emerge from Fig. 6: i) a less fresh North Atlantic north of $65°$ N (Nordic Seas) in E3SM-Arctic-OSI compared to E3SM-LR-OSI, also associated with a deeper reaching convection in the same area (see missing or greatly reduced slumping of sigma2 isopycnals between $65°$ N and $75°$ N in the E3SM-LR third panel); ii) a steeper Southern Ocean stratification in E3SM-Arctic-OSI, which causes Circumpolar Deep Waters associated with sigma2 of $36.8 - 37$ kg m$^{-3}$ to remain well below the surface in E3SM-LR-OSI. Both of these features are consistent with the presence of a stronger AMOC in E3SM-Arctic compared to E3SM-LR and, partially, with the results of Bryan et al. (2014).

While we acknowledge the global biases discussed in this section, we also note that they fall within the published inter-model spread of results from forced climate models (Danabasoglu et al., 2014; Tsujino et al., 2020). Furthermore, as we discuss in the next sections, the E3SM-Arctic-OSI simulation of the Arctic is satisfactory, and improvements in MPAS-Ocean and MPAS-Seaice introduced in E3SMv2 are expected to yield further improvements in future E3SM-Arctic simulations of global and high-latitude climate.

## 4 Arctic Gateways

Given the importance of the Arctic-Subarctic ocean exchanges to regional and global climate change, we examine the multi-year mean simulated ocean fluxes across the five main gateways that connect the Arctic Ocean with the subpolar North Atlantic and North Pacific Oceans (see Beszczynska-Möller et al., 2011, for an overview). Warm and salty water of Atlantic origin enters the Arctic through the BSO and Fram Strait, while warm and fresh water of Pacific origin flows into the Arctic through the Bering Strait. The outflow of water from the Arctic takes place through Nares Strait and Northwest Passages in the Canadian Archipelago (and eventually Davis Strait), and through Fram Strait. Observational values of the volume transport (and heat and freshwater transports, when available) through these gateways vary according to the time period over which the observations were actually taken. Typically cited numbers for the volume transports include a net inflow of $2 \pm 0.6$ Sv through the BSO (based on measurements between 1997 and 2007, from Skagseth et al. (2008) and Smedsrud et al. (2013)); a net inflow through Bering Strait of $0.8 \pm 0.2$ Sv (1990-2007, from Woodgate and Aagaard, 2005); a net outflow through Fram Strait of $2 \pm 2.7$ Sv (1997-2007, from Schauer et al., 2008); and a net outflow through Davis Strait of $1.6 \pm 0.5$ Sv (2004-2010, from Curry et al., 2014). Observations are also available in key channels of the Canadian Archipelago, such as Lancaster Sound and Nares Strait (Fig. 1c), but they are over shorter time records (see captions of subsequent figures and Table 2 for references to specific studies).

In the remainder of this section we focus on how the model reproduces exchanges of volume, heat, and liquid freshwater through the above mentioned Arctic gateways. Full time series of the net fluxes are presented in Figs. 7-9, and mean values are summarized in Table 2). They are compared with available observations and with the multi-model studies of Ilicak et al. (2016) and Wang et al. (2016a). It should be noted that integrated long-term observational volume flux estimates yield an imbalance of 0.8 Sv (per Table 2), while the respective model integrated volume estimates are by definition close to zero ($< 0.1$ Sv), which complicates comparison of fluxes at individual gates. In addition, a full volume, heat, and freshwater budget of the Arctic is beyond the scope of this paper. Finally, we acknowledge that estimates of heat and freshwater fluxes across an individual

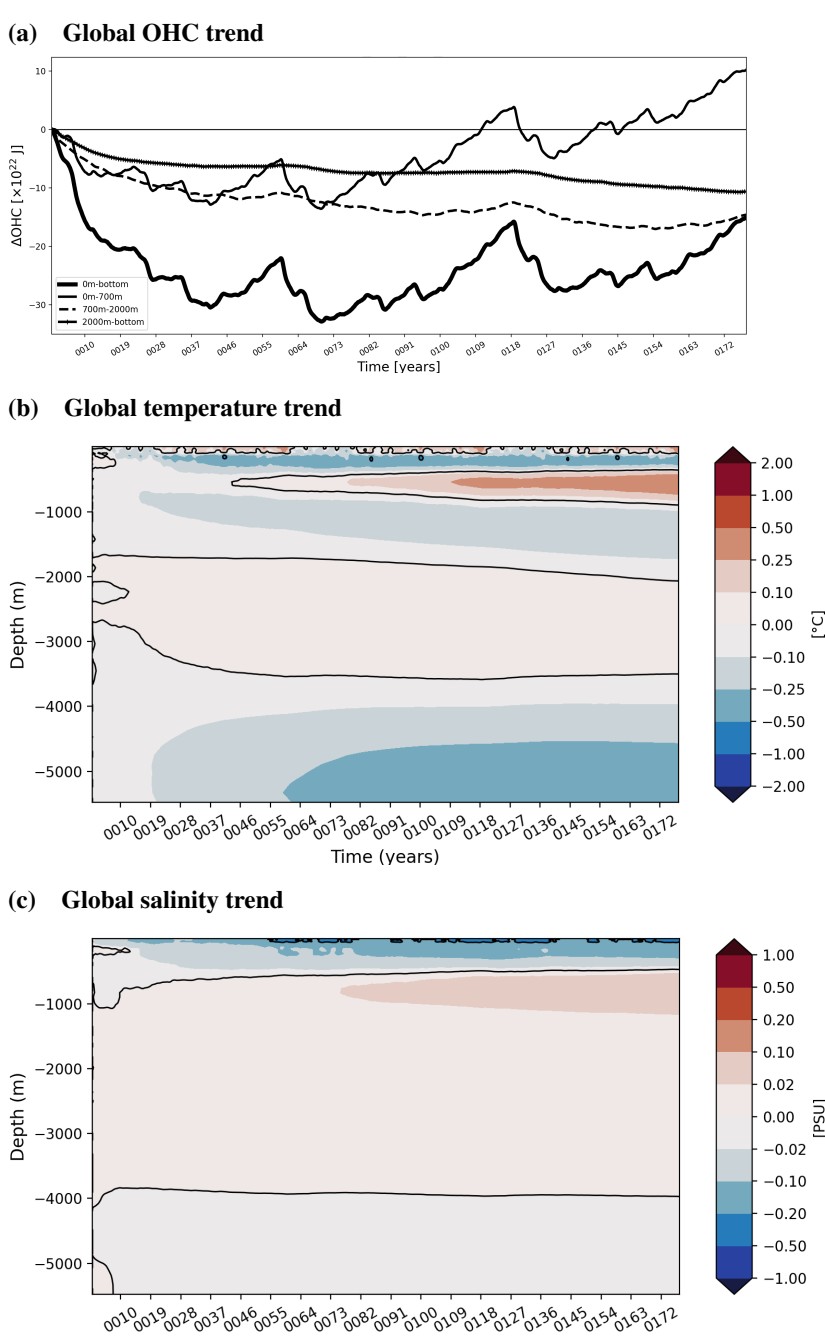

**Figure 2.** Global trends for the E3SM-Arctic-OSI simulation of (a) ocean heat content (OHC) anomalies integrated over the full depth column (thick solid line), and over the following depth ranges: $0 - 700$ m (thin solid line), $700 - 2000$ m (dashed line), and $2000$ m$-$bottom (plus line); (b) temperature and (c) salinity anomalies as a function of depth. Anomalies are computed with respect to the first-year annual mean, and are 1-year running averaged.

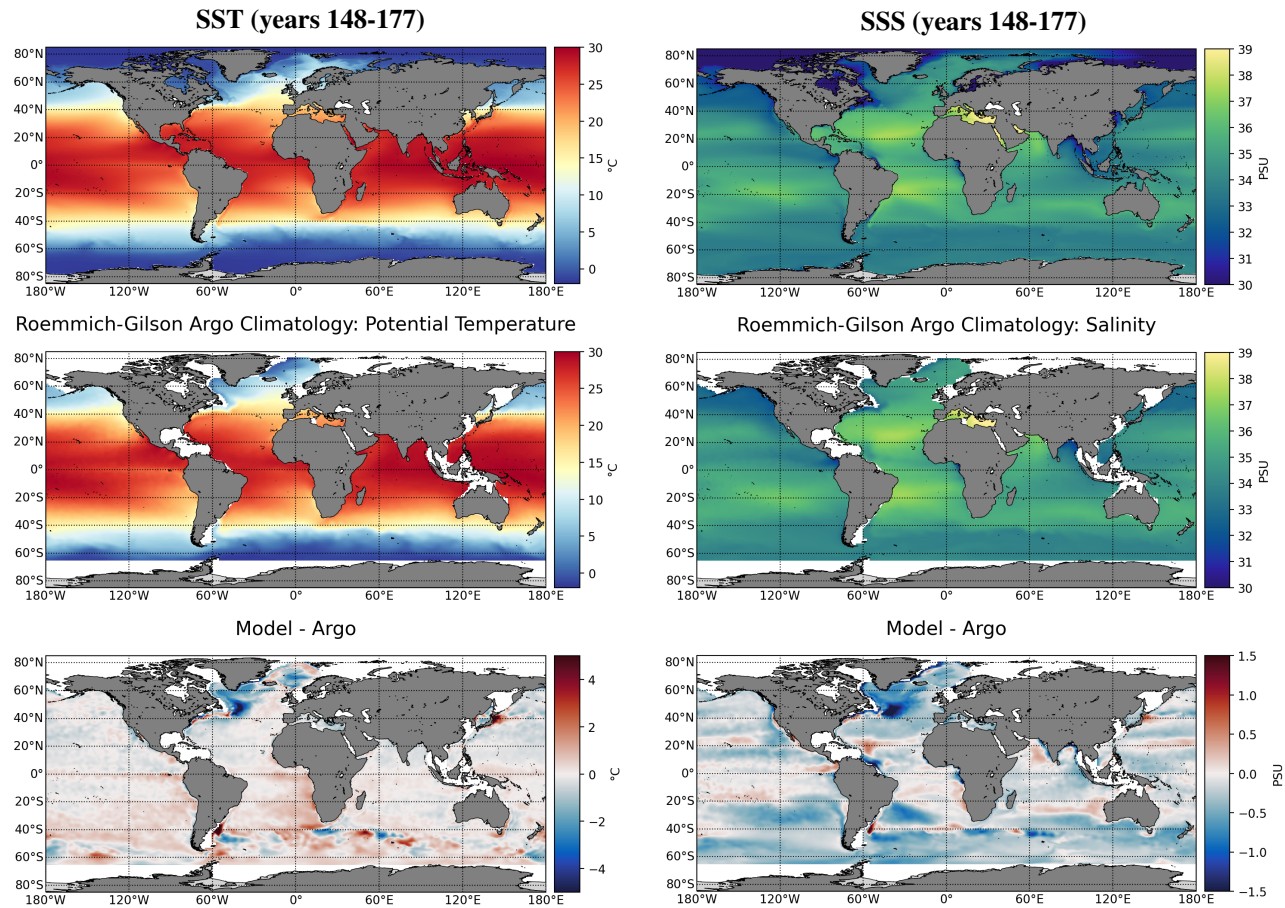

**Figure 3.** SST (left panels) and SSS (right panels) from the E3SM-Arctic-OSI simulation (upper) and the Roemmich-Gilson Argo climatological data set (middle). The corresponding model minus observation bias is shown in the lower panels. Model climatologies are computed over years 148-177 (last 30 years of the third JRA cycle).

section are sensitive to the respective reference values used for temperature and salinity, but we argue that their accumulated quantities for a closed volume are justified. For the heat transport calculations, either the freezing point or $0°C$ is used as the reference temperature, depending on the reference used by the corresponding observational estimate (where available); for the freshwater transport, $34.8$ psu is used as the reference salinity, since that is the common reference used in Arctic observational

studies. The sign convention for these transports is such that positive values imply net fluxes into the Arctic Ocean, and negative values imply net fluxes out of the Arctic.

Fram Strait and the BSO are two important gateways both in terms of heat and freshwater transport into and out of the Arctic: the E3SM-Arctic-OSI simulation reproduces well the mean volume, heat, and freshwater transports through these gateways compared to observations (red lines in Figs. 7a,b, 8a,b, 9a,b, and mean values in first 2 rows of Table 2). While the

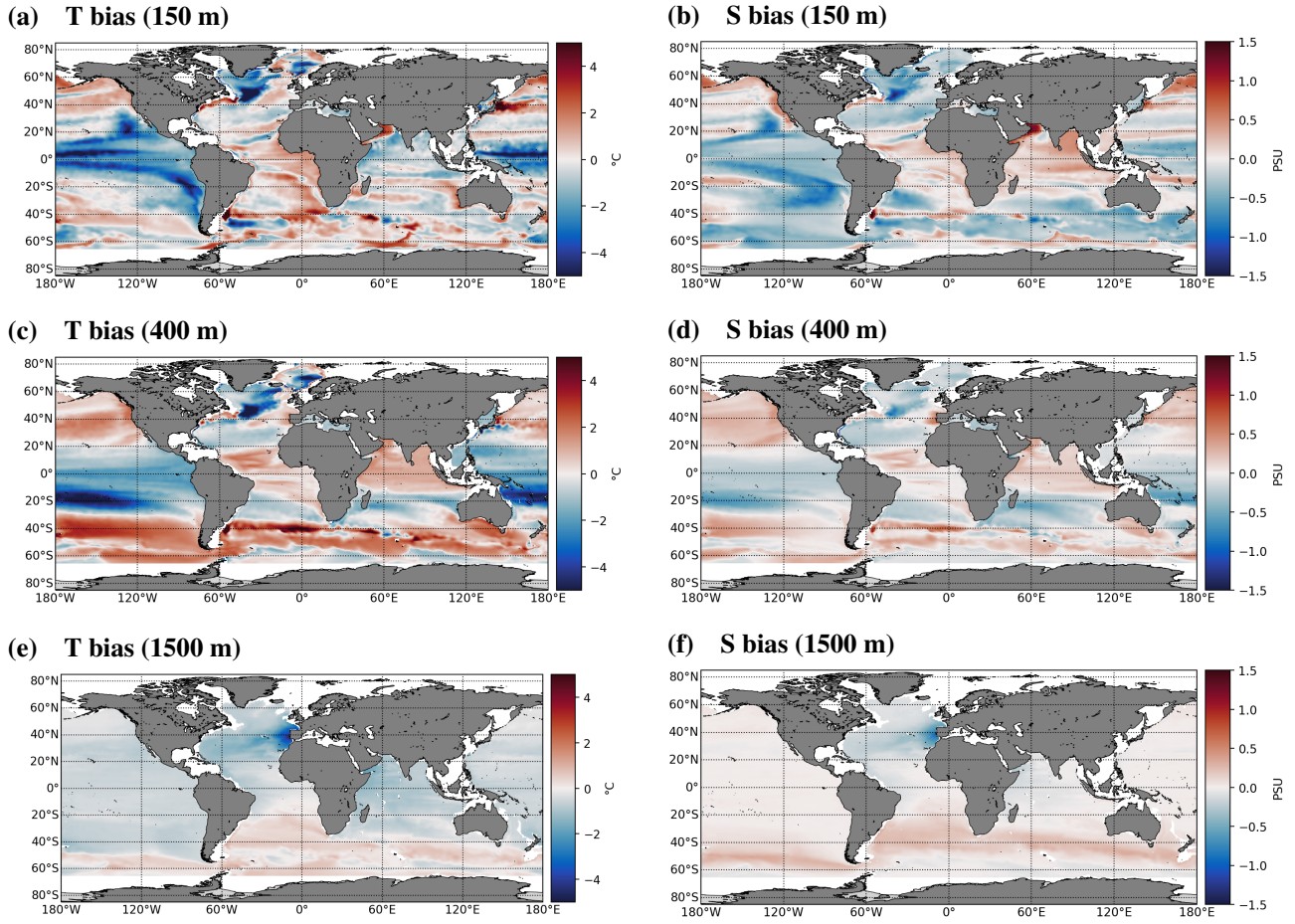

**Figure 4.** Model minus observation bias for (a,c,e) temperature and (b,d,f) salinity at depths of (a,b) 150 m, (c,d) 400 m, and (e,f) 1500 m. Biases are computed similarly to the lower panels of Fig. 3 and for model climatologies over years 148-177.

net volume transport through these two gateways is not characterized by an appreciable trend over each JRA cycle, the net heat transport through both Fram Strait and the Barents Sea exhibits an upward trend over the last $\approx 40$ years of each cycle (Fig. 8a,b), and the net freshwater transport through the BSO exhibits a downward trend over the same time period (Fig. 9b; note that less negative freshwater flux effectively means that the Arctic is losing less freshwater with respect to $34.8$ psu through the BSO). These results are in agreement with observational studies such as Skagseth et al. (2008); Schauer et al. (2008); Polyakov et al. (2017), promoting the idea of 'Atlantification' of the Arctic, with warmer and saltier Atlantic water flowing into the Arctic in recent decades. The simulated net freshwater transport through Fram Strait is more variable and follows quite closely the observational record of the Norwegian Polar Institute (de Steur et al., 2009; de Steur, 2018, see also graph at http://www.mosj.no/en/climate/ocean/freshwater-flux-fram-strait.html). Net volume fluxes through Fram Strait and the BSO are also well reproduced in E3SM-LR-OSI, but net heat transport is $\sim 4$ times weaker through Fram Strait and 38% weaker

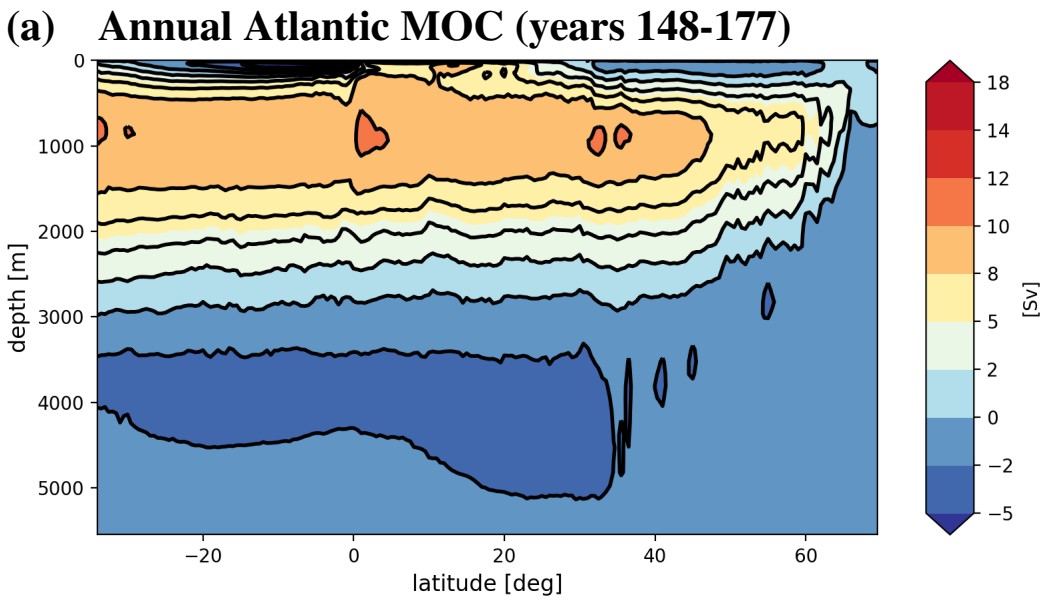

**(a)  Annual Atlantic MOC (years 148-177)**

**(b)  Max Atlantic MOC at 26.5N**

**Figure 5.** (a) Annual MOC streamfunction computed over the Atlantic Ocean and over years 148-177 of the E3SM-Arctic-OSI simulation (black contours are every 2 Sv); (b) time series of the 5-year running average maximum Atlantic MOC detected at $26.5°$ N (latitude of the RAPID-MOCHA observational array) from E3SM-Arctic-OSI (dark red line) and E3SM-LR-OSI (black line). The light red line shows E3SM-Arctic-OSI monthly values. The numbers shown in the insets are the mean and standard deviations of the annual model values, computed over the full time series. The RAPID array typical variability ($16.8 \pm 4.4$ Sv) is shaded in green. Finally, the purple vertical lines show the transition across JRA cycles.

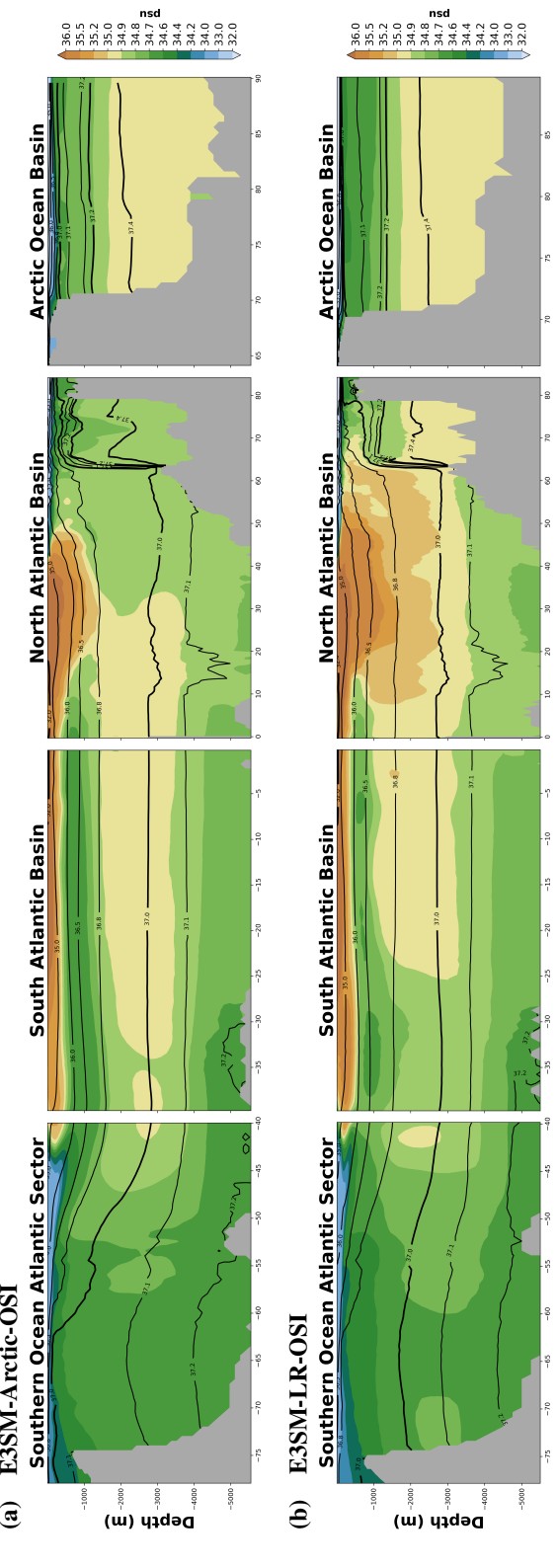

**Figure 6.** Meridional vertical section of zonally averaged salinity field, where the zonal average is computed over the Atlantic sector of the global ocean and over the whole Arctic Ocean north of approximately the latitude of Fram Strait, from (a) E3SM-Arctic-OSI and (b) E3SM-LR-OSI. Contour lines follow the sigma2 field (potential density with respect to 2000 m). All fields are climatologies over years 148-177.

through the BSO than E3SM-Arctic-OSI. Fram Strait is also characterized by an almost twice as intense net freshwater export with respect to the E3SM-Arctic-OSI results (and observations; see black lines in Fig. 9a and Table 2).

Lancaster Sound and Nares Strait (Fig. 1c) are the only connections from the Arctic to Baffin Bay through the Canadian Archipelago, since both Cardigan Strait and Hell's Gate to the north of Lancaster Sound are closed in our E3SM-Arctic-OSI configuration. On average, the net volume transports through these two gateways compare very well with available observations, and their sum defines the mean volume transport through Davis Strait (Fig. 7d,e,f). While net heat transport through Lancaster Sound and Nares Strait is very small compared to the other gateways, heat flux through Davis Strait is almost one order of magnitude higher ($\approx 10$ TW on average), suggesting that there is a substantial heat loss to the atmosphere in Baffin Bay. The Canadian Archipelago gateways are most important for the transport of freshwater out of the Arctic (Fig. 9d,e). Similarly to Fram Strait, the freshwater transport through Lancaster Sound and Nares Strait exhibits substantial interannual and decadal variability, something that is not fully captured by observations, likely due to the limited coverage of these records (observational range is based on the 1998-2001 record in Lancaster Sound from Prinsenberg and Hamilton (2005) and on the 2003-2009 record in Nares Strait from Münchow (2016)). Because of the low horizontal resolution and the fact that Nares Strait is closed in E3SM-LR-OSI, net volume transport through Lancaster Sound is not significantly different from 0 and all net fluxes through Davis Strait are much reduced (by up to three times) in E3SM-LR-OSI compared with E3SM-Arctic-OSI results and observations. Note from Table 2 that this reduction of outward volume transport through Davis Strait in E3SM-LR-OSI with respect to E3SM-Arctic-OSI, is partly compensated by an increase in transport through Fram Strait but also by a decrease in transport into the Arctic through the BSO and Bering Strait.

The two simulations interestingly reproduce the net fluxes through Bering Strait similarly (panel (c) in Figs. 7-9), with volume and freshwater transports well represented compared with observations, and net heat transport on the lower end of observational estimates as also found in other ESM studies (Ilicak et al., 2016). Furthermore, we note a downward trend in the net freshwater flux over the last 20 years of each JRA cycle (corresponding to the period $\sim 2006 - 2016$; Fig. 9c), which is opposite to the observed upward trend reported in Woodgate (2018).

While the net transports through the Arctic gateways are useful diagnostics, it is equally important for a model to reproduce inflows and outflows, since those are associated with different water masses and their impacts downstream are commonly independent from each other. Figs. 10-13 show vertical sections from both E3SM-Arctic-OSI and E3SM-LR-OSI of potential temperature, salinity, and normal velocity for Fram Strait, the BSO, Davis Strait, and Bering Strait, respectively (Table 2 also include the averaged values of incoming and outgoing transports for all fluxes). The model climatologies are computed over the last 12 years of the third JRA cycle. A comparison with climatologies computed on an analogous period of the first cycle (not shown) indicate that, while some T and S changes are apparent below the Atlantic Water layer in Fram Strait and the BSO, and in the West Greenland Current in Davis Strait, the overall structure of the gateways stratification is quite consistent between the first and third JRA cycle, and consequently the velocity structures are also very comparable.

Fram Strait results in terms of the cross-section of temperature and normal velocity (Fig. 10a,e) are compared with the 2002-2008 observational climatologies in Beszczynska-Möller et al. (2012, their Fig. 2), whereas the salinity cross-section (Fig. 10c) can be compared with 1997 observations in Rudels (2012, his Fig. 20). They show a good representation of the currents and

exchanges through the strait. In particular, the West Spitsbergen Current carrying warm and salty modified Atlantic Water into the Arctic west of the Svalbard Islands, besides being weaker in its core and warmer by $\approx 1°C$ in the western side of the section compared with observations, exhibits a good vertical structure as well as reasonable temperature and salinity ranges. The same is true for the East Greenland Current carrying cold and fresh polar waters out of the Arctic along the Greenland shelf. Note that the West Spitsbergen Current carries slightly less water into the Arctic than what the East Greenland Current carries out of the Arctic, but the former is responsible for a net input of heat and the latter is responsible for a net loss of freshwater for the Arctic Ocean (see Table 2).

The Fram Strait stratification and cross-section velocity in E3SM-LR-OSI looks very different: the upper 500 m of the water column is much more stratified than E3SM-Arctic-OSI, exhibiting a very fresh and cold lens in the top 100 m that maybe associated with an excessive sea-ice export out of the Arctic. These temperature and salinity profiles in E3SM-LR-OSI may also explain the excessive net freshwater flux and reduced net heat flux through Fram Strait discussed above.

Since the bulk of the water flowing across the BSO has salinities greater than the reference salinity of 34.8 psu (Fig. 11d), this represents freshwater export from the Arctic (see negative values for both net and incoming freshwater for the BSO in Table 2). Compared with observations (Fig. 3 in Skagseth et al., 2008, which in truth only represent conditions for August 1998, while the model results are interannual climatologies), the model Atlantic water flowing into the Arctic through the BSO is slightly fresher and colder, but the slope and interior currents are well simulated in terms of both horizontal and vertical structure (Fig. 11a,c,e). The corresponding results for E3SM-LR-OSI (Fig. 11b,d,f) are also acceptable, although they exhibit a weaker Norwegian Atlantic Current than in E3SM-Arctic-OSI and observations.

The Davis Strait temperature and salinity cross-sections are also well simulated in E3SM-Arctic-OSI (the results, seen in Fig. 12a,c, are compared with Tang et al. (2004), their Fig. 4), whereas a strong stratification caused by a low-salinity upper 150 m layer is present in E3SM-LR-OSI. On the other hand, and as mentioned earlier in this section, both simulations represent the inflow of Pacific water through the Bering Strait (Fig. 13) in a similar fashion, with an underestimation of the incoming Pacific water temperature by several degrees compared with observations presented in Woodgate et al. (2015) and the model results discussed in Clement Kinney et al. (2014).

## 5  Arctic Ocean and Sea-Ice Conditions

In this section, we characterize the E3SM-Arctic-OSI simulated ocean and sea-ice conditions in the central Arctic, focusing on the following metrics: ocean stratification, fresh water content, and sea-ice concentration and thickness. We consider both the trends and climatologies of these quantities, and compare results with E3SM-LR-OSI, the RASM model (see section 2 for a description of the RASM simulation used here), and observations when available.

### 5.1  Ocean hydrology and freshwater content

Seasonal (Jan-Feb-Mar, or JFM, and July-Aug-Sep, or JAS) hydrographic profiles of the E3SM-Arctic-OSI and E3SM-LR-OSI simulations are computed over the Arctic Ocean and over the region of the Canada Basin, and compared with i) RASM model

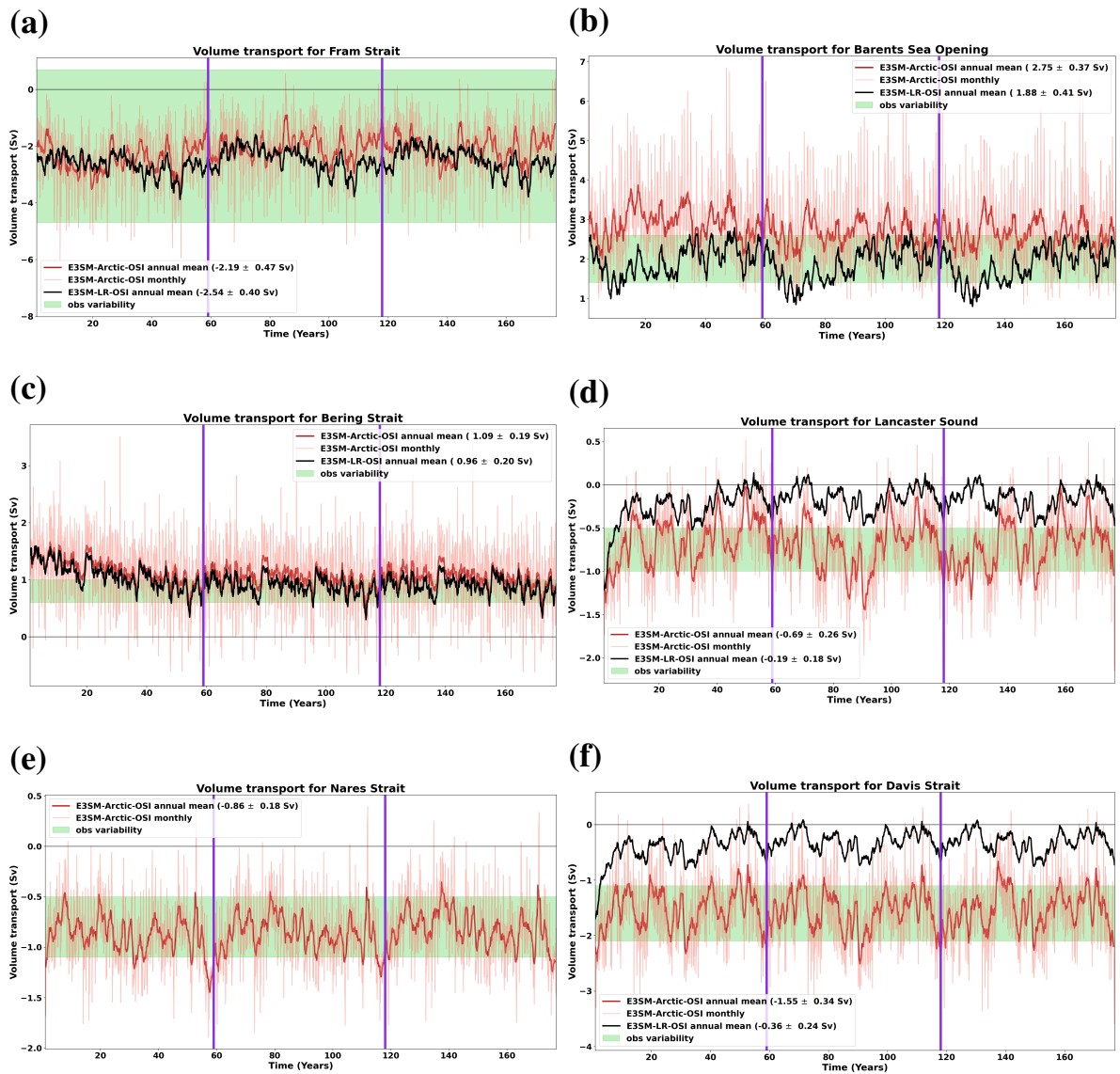

**Figure 7.** Time series of the 1-year running average net volume transports for the five Arctic gateways (Fram Strait, BSO, Bering Strait, Lancaster Sound, and Nares Strait) and Davis Strait, from E3SM-Arctic-OSI (dark red lines) and E3SM-LR-OSI (black lines). The light red lines show E3SM-Arctic-OSI monthly values; the corresponding E3SM-LR-OSI monthly values are not shown for clarity. The vertical purple lines mark the transition between JRA cycles. The numbers shown in the insets are the mean and standard deviations of the annual model values, computed over the full time series. Transect location is displayed in Fig. 1c. Observational values are $-2 \pm 2.7$ Sv for Fram Strait (Schauer et al., 2008), $2 \pm 0.6$ Sv for the BSO (Skagseth et al., 2008; Smedsrud et al., 2013), $0.8 \pm 0.2$ Sv for Bering Strait (Woodgate and Aagaard, 2005), $-0.75 \pm 0.25$ Sv for Lancaster Sound (Prinsenberg and Hamilton, 2005), between $-0.5$ and $-1.1$ Sv for Nares Strait (Münchow, 2016), and $-1.6 \pm 0.5$ Sv for Davis Strait (Curry et al., 2014).

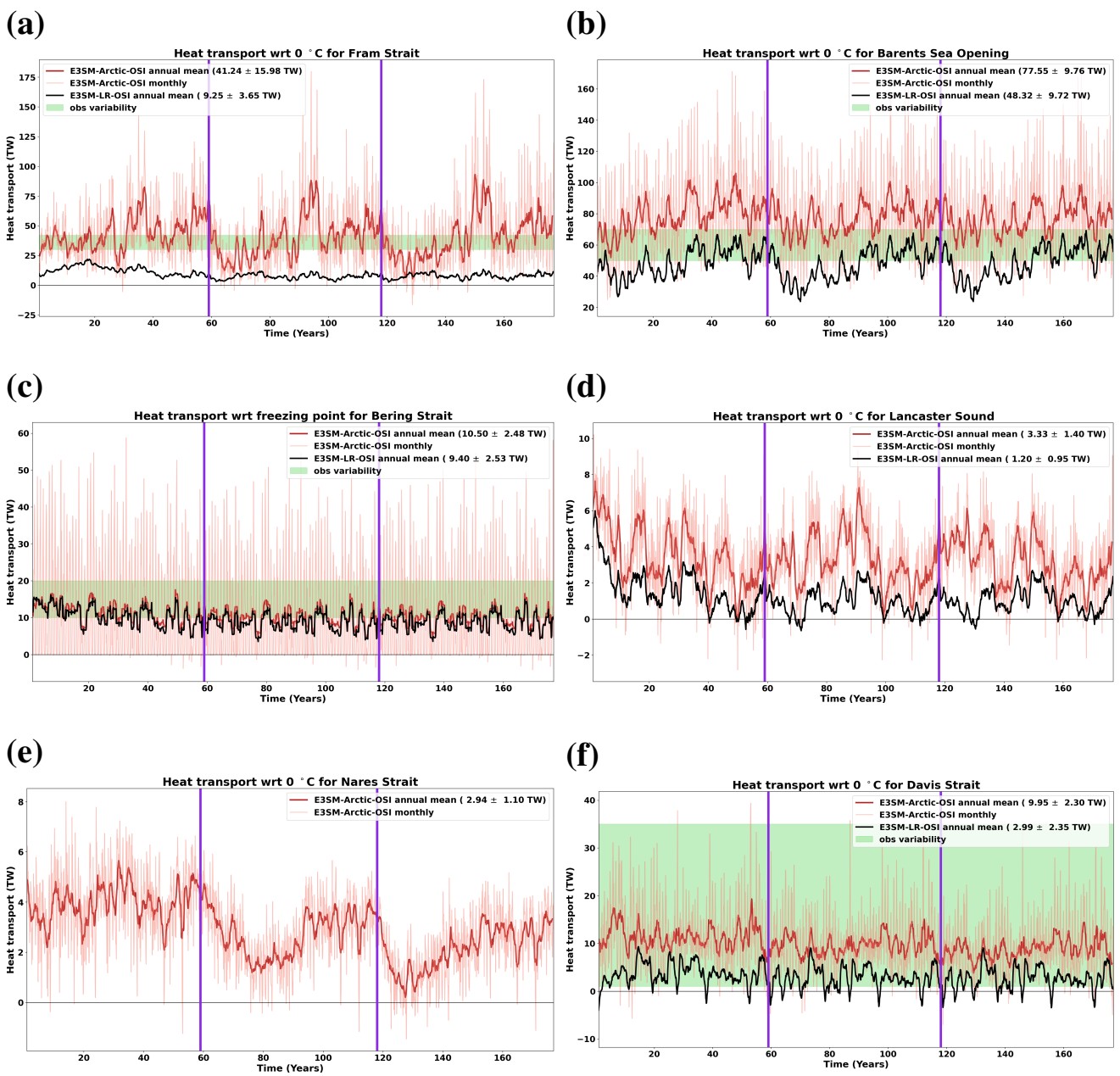

**Figure 8.** Similar to Fig. 7, but for net heat transport, which is computed with respect to the $0°C$ reference temperature in panels (a),(b),(d)-(f), and with respect to the freezing point in panel (c). Observational ranges, where available, are shaded in green; their values are $36 \pm 6$ TW for Fram Strait (Schauer and Beszczynska-Möller, 2009), between 50 and 70 TW for the BSO (Smedsrud et al., 2010), between 10 and 20 TW for Bering Strait (Woodgate et al., 2010, these authors use the freezing point as reference temperature, as done for the model estimates), and $18 \pm 17$ TW for Davis Strait (Cuny et al., 2005).

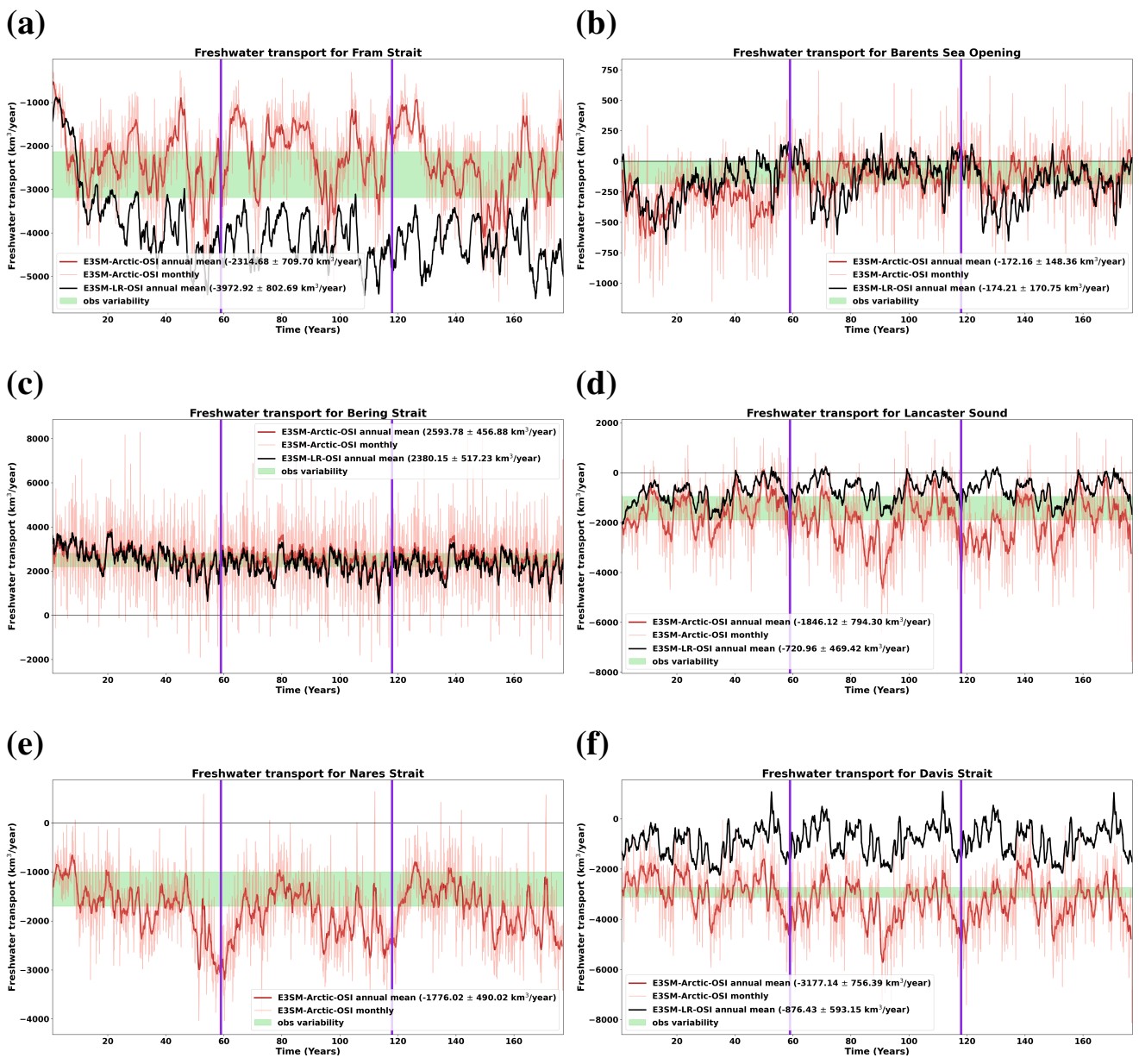

**Figure 9.** Similar to Fig. 7, but for net freshwater transport, where freshwater is computed with respect to the reference salinity of 34.8 psu. Observational values are $-2660 \pm 528$ km³ year⁻¹ for Fram Strait and $-90 \pm 94$ km³ year⁻¹ for the BSO (see Serreze et al., 2006, for both estimates), $2500 \pm 300$ km³ year⁻¹ for Bering Strait (Woodgate and Aagaard, 2005), between $-1900$ and $-950$ km³ year⁻¹ for Lancaster Sound (Prinsenberg and Hamilton, 2005), between $-1700$ and $-1000$ km³ year⁻¹ for Nares Strait (Münchow, 2016), and $-2930 \pm 190$ km³ year⁻¹ for Davis Strait (Curry et al., 2014).

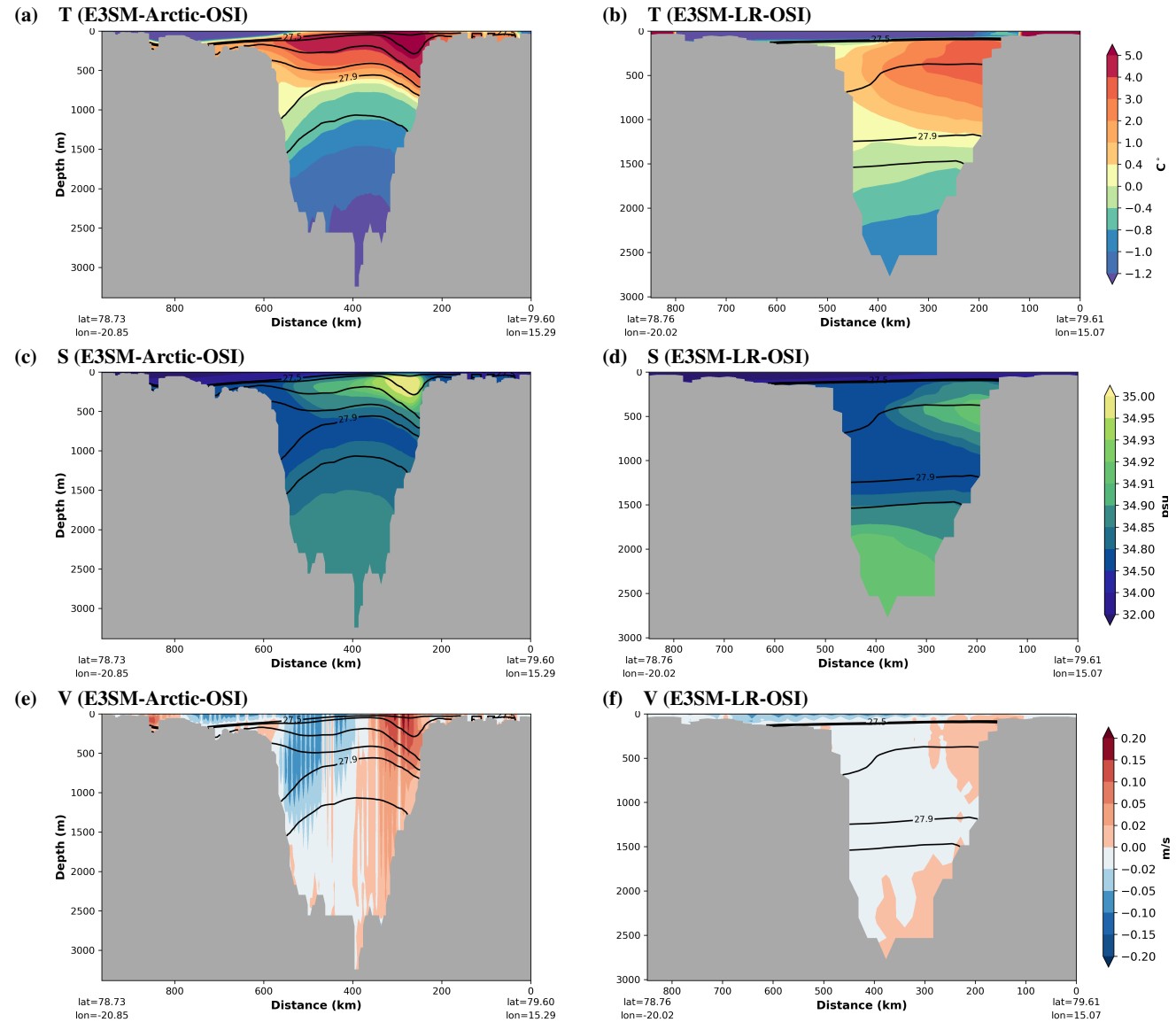

**Figure 10.** Cross-section of (a,b) potential temperature, (c,d) salinity, and (e,f) normal velocity for Fram Strait, for (a,c,e) E3SM-Arctic-OSI and (b,d,f) E3SM-LR-OSI (note that these fields are plotted on the native MPAS mesh, identifying the mesh cells that fall onto the specific transect; this is the reason for the noisy velocity values in (e,f)). Annual climatologies are computed over years 166-177. Black contours show potential density (sigma0).

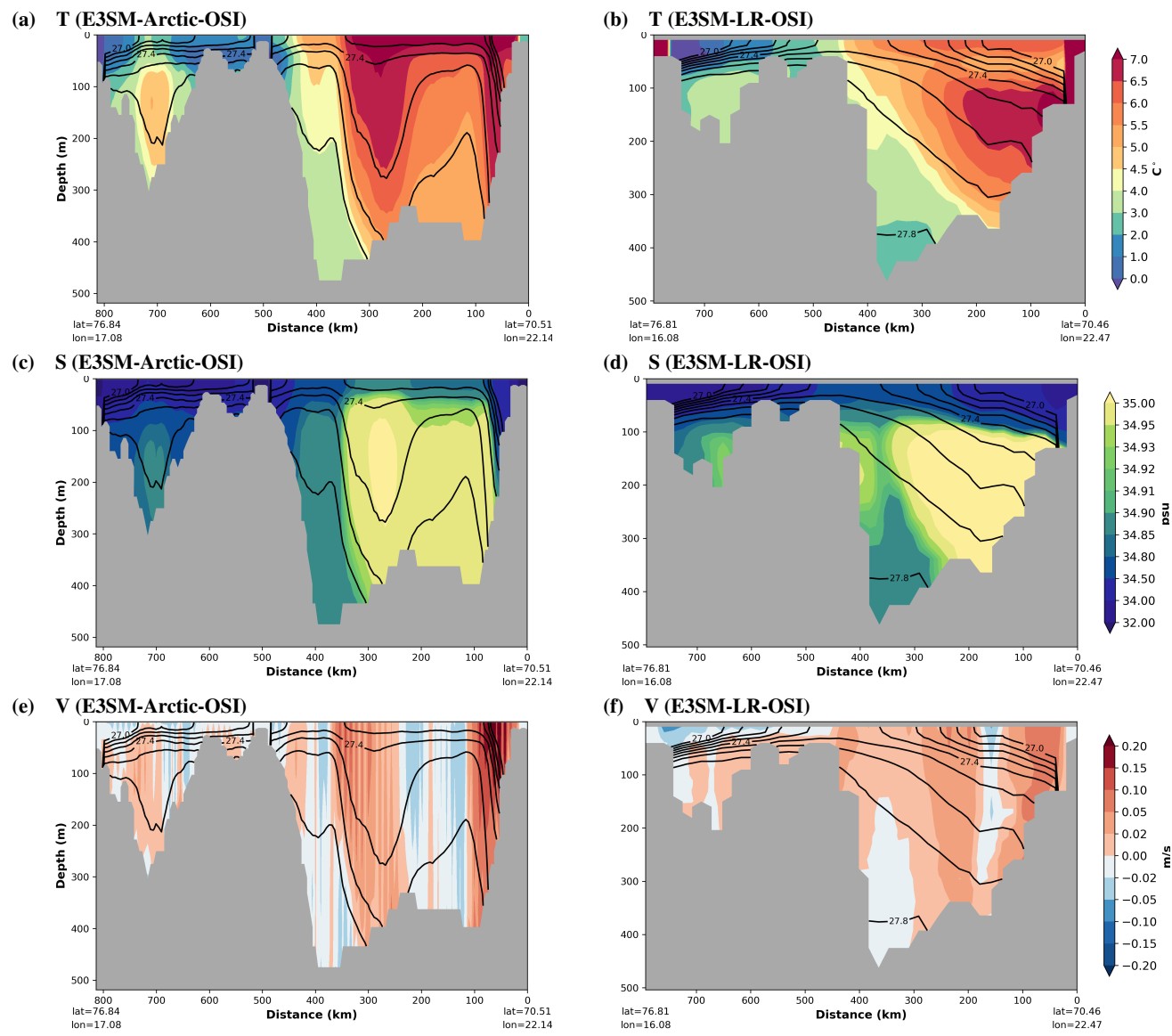

**Figure 11.** Similar to Fig. 10, but for the Barents Sea Opening.

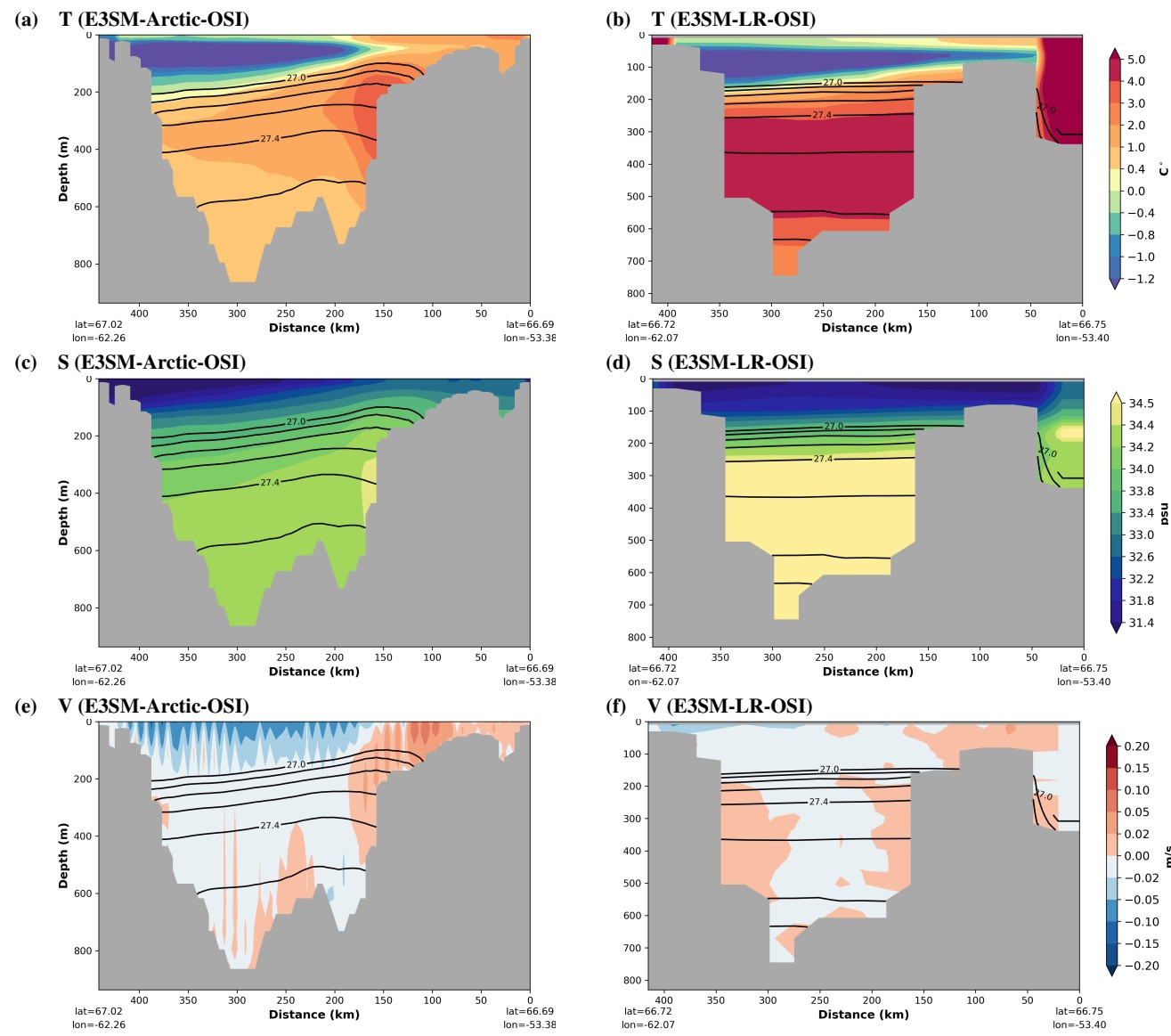

**Figure 12.** Similar to Fig. 10, but for Davis Strait.

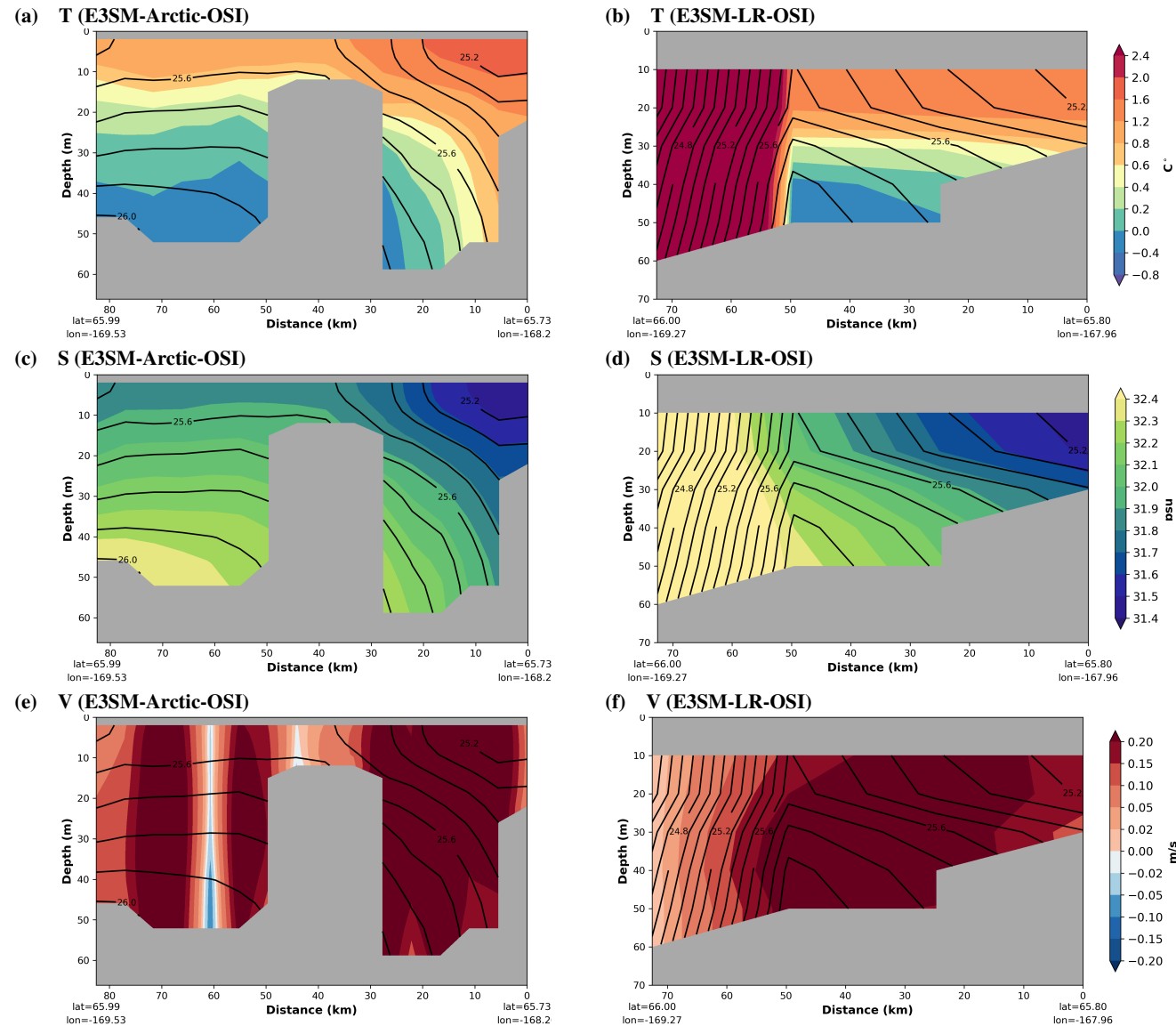

**Figure 13.** Similar to Fig. 10, but for Bering Strait.

**Table 2.** Mean fluxes and variability through Arctic gateways from E3SM-Arctic-OSI, E3SM-LR-OSI, RASM, and available observations. Note that heat flux is referred to 0°C everywhere except for the Bering Sea where freezing temperature is used as reference (indicated by $T_{FP}$). The observational heat flux value for Fram Strait is computed for closed volume budget (reference temperature is arbitrary in this case). Finally, freshwater flux is referred to a 34.8 psu salinity.

| Transect | | Volume net (Sv) | Volume in | Volume out | Heat net (TW) | Heat in | Heat out | FW net (km³/year) | FW in | FW out |
|---|---|---|---|---|---|---|---|---|---|---|
| Fram Strait | E3SM-Arctic | −2.18 ± 1.03 | 10.82 ± 2.53 | −13.00 ± 2.47 | 41.38 ± 26.15 | 86.76 ± 36.22 | −45.38 ± 25.20 | −2312.20 ± 917.51 | 408.95 ± 669.30 | −2721.16 ± 1139.92 |
| | E3SM-LR | −2.54 ± 0.92 | 1.21 ± 0.55 | −3.75 ± 1.09 | 9.27 ± 5.39 | 3.02 ± 3.47 | 6.26 ± 2.76 | −3969.56 ± 1026.87 | 278.17 ± 283.45 | −4247.73 ± 1085.08 |
| | RASM | −2.06 ± 1.14 | 4.91 ± 1.31 | −6.97 ± 1.70 | 29.86 ± 13.70 | 53.02 ± 19.94 | −23.16 ± 10.98 | −1646.48 ± 644.35 | 1554.23 ± 431.22 | −3200.71 ± 854.43 |
| | obs | −2 ± 2.7[a] | – | – | 36 ± 6[b] | – | – | −2660 ± 528[c] | – | – |
| BSO | E3SM-Arctic | 2.74 ± 0.95 | 5.02 ± 0.99 | −2.28 ± 0.51 | 77.45 ± 24.61 | 111.91 ± 28.11 | −34.46 ± 9.95 | −170.94 ± 251.29 | −156.66 ± 332.92 | −14.28 ± 196.95 |
| | E3SM-LR | 1.88 ± 0.86 | 2.76 ± 0.84 | −0.88 ± 0.27 | 48.29 ± 17.23 | 55.81 ± 18.52 | −7.52 ± 3.37 | −173.42 ± 270.19 | 192.00 ± 202.58 | −365.42 ± 229.94 |
| | RASM | 3.13 ± 1.05 | 4.42 ± 1.11 | −1.29 ± 0.34 | 73.78 ± 25.07 | 91.95 ± 26.94 | −18.16 ± 6.71 | −213.07 ± 348.23 | 598.66 ± 252.15 | −811.74 ± 277.81 |
| | obs | 2 ± 0.6[d] | – | – | 50 to 70[e] | – | – | −90 ± 94[c] | – | – |
| Bering Strait | E3SM-Arctic | 1.09 ± 0.57 | 1.16 ± 0.56 | −0.07 ± 0.06 | 10.50 ± 13.03 | 11.05 ± 13.46 | −0.55 ± 0.72 | 2594.04 ± 1443.30 | 2756.88 ± 1386.49 | −162.84 ± 203.93 |
| | E3SM-LR | 0.96 ± 0.57 | 0.97 ± 0.54 | −0.02 ± 0.08 | 9.40 ± 12.35 | 9.48 ± 12.28 | −0.08 ± 0.56 | 2382.52 ± 1511.45 | 2431.69 ± 1413.56 | −49.17 ± 229.62 |
| | RASM | 0.70 ± 0.35 | 0.71 ± 0.33 | −0.01 ± 0.04 | 5.62 ± 8.05 | 5.84 ± 7.88 | −0.23 ± 0.82 | 2065.65 ± 1111.91 | 2096.08 ± 1052.34 | −30.43 ± 138.37 |
| | obs | 0.8 ± 0.2[f] | – | – | 10 to 20[g] ($T_{FP}$) | – | – | 2500 ± 300[f] | – | – |
| Lancaster Sound | E3SM-Arctic | −0.69 ± 0.40 | 0.06 ± 0.05 | −0.75 ± 0.38 | 3.35 ± 2.00 | −0.20 ± 0.26 | 3.55 ± 1.89 | −1853.72 ± 1205.13 | 121.09 ± 168.49 | −1974.81 ± 1129.99 |
| | E3SM-LR | −0.20 ± 0.27 | 0.04 ± 0.08 | −0.24 ± 0.23 | 1.21 ± 1.34 | −0.13 ± 0.32 | 1.33 ± 1.18 | −724.46 ± 746.52 | 79.49 ± 190.87 | −803.94 ± 639.74 |
| | RASM | −0.83 ± 0.27 | 0.01 ± 0.01 | −0.84 ± 0.27 | 4.01 ± 1.43 | 4.03 ± 1.42 | −0.02 ± 0.05 | −2043.40 ± 774.79 | 12.74 ± 55.21 | −2056.14 ± 758.35 |
| | obs | −0.75 ± 0.25[h] | – | – | – | – | – | −1900 to −950[h] | – | – |
| Nares Strait | E3SM-Arctic | −0.86 ± 0.32 | 0.01 ± 0.03 | −0.87 ± 0.31 | 2.94 ± 1.51 | −0.01 ± 0.09 | 2.95 ± 1.48 | −1777.92 ± 710.98 | 7.72 ± 44.81 | −1785.64 ± 695.39 |
| | E3SM-LR | – | – | – | – | – | – | – | – | – |
| | RASM | −0.90 ± 0.21 | 0.00 ± 0.00 | −0.90 ± 0.21 | 3.24 ± 1.00 | 3.26 ± 1.00 | −0.01 ± 0.03 | −1221.94 ± 330.51 | 0.32 ± 4.39 | −1222.26 ± 329.47 |
| | obs | −1.1 to −0.5[i] | – | – | – | – | – | −1700 to −1000[i] | – | – |
| Davis Strait | E3SM-Arctic | −1.55 ± 0.60 | 1.25 ± 0.55 | −2.80 ± 0.54 | 9.93 ± 6.27 | 6.28 ± 5.67 | 3.65 ± 4.41 | −3181.77 ± 1112.25 | 1616.74 ± 869.82 | −4798.51 ± 1264.05 |
| | E3SM-LR | −0.36 ± 0.36 | 0.46 ± 0.26 | −0.82 ± 0.32 | 2.93 ± 5.64 | 3.13 ± 2.84 | −0.20 ± 3.92 | −875.43 ± 1066.37 | 726.65 ± 667.77 | −1602.09 ± 784.86 |
| | RASM | −1.75 ± 0.47 | 0.94 ± 0.58 | −2.69 ± 0.47 | 9.35 ± 4.91 | 13.73 ± 5.87 | −4.38 ± 3.25 | −3474.15 ± 933.41 | 890.69 ± 582.83 | −4364.84 ± 1091.41 |
| | obs | −1.6 ± 0.5[j] | – | – | 18 ± 17[k] | – | – | −2930 ± 190[j] | – | – |

[a] Schauer et al. (2008)
[b] Schauer and Beszczynska-Möller (2009)
[c] Serreze et al. (2006)
[d] Skagseth et al. (2008); Smedsrud et al. (2013)
[e] Smedsrud et al. (2010)
[f] Woodgate and Aagaard (2005)
[g] Woodgate et al. (2010)
[h] Prinsenberg and Hamilton (2005)
[i] Münchow (2016)
[j] Curry et al. (2014)
[k] Cuny et al. (2005)

results (climatologies computed over years 2005-2014); ii) Ice-Tethered Profiler (ITP) observations[2] (Toole et al., 2011); and iii) the World Ocean Atlas 2018 climatology (WOA18; Locarnini et al., 2018; Zweng et al., 2018). E3SM-Arctic-OSI model results are averaged over two different periods of the third JRA cycle, years 125-149, corresponding to 1964-1988, and years

166-177, corresponding to 2005-2016, so as to characterize the early and late periods identified by the purple bars in Fig. 17 (see section 5.2); E3SM-LR-OSI results are instead shown for the later period only (Figures 14-15). In addition, E3SM-Arctic-OSI results are also shown for years 48-59 (end of first JRA cycle), in order to compare the stratification from the third and first cycles (solid and dashed dark red lines in Figs 14-15). Density profiles (not shown) resemble closely the salinity profiles in the upper 800 m of the Arctic water column. A general feature that can be noted is that both E3SM simulations predict a fresher

overall Arctic in the upper 150 m compared with WOA climatology and RASM results, especially in the JFM season (Figs 14d, 15d). This fresh bias is reduced by approximately half when going from E3SM-LR-OSI to E3SM-Arctic-OSI (bias changes from $1-3$ psu to $0.7-2$ psu, respectively). The shape of the overall Arctic thermocline is well reproduced in E3SM-Arctic-OSI, whereas both E3SM-LR-OSI and RASM predict a more smoothed temperature profile below 300 m; having said that, E3SM-Arctic-OSI tends to overestimate temperature by up to $\approx 1.5°C$ in the $100-800$ m depth range (Figs. 14a,b, 15a,b),

which corresponds with the Atlantic water layer. Looking more closely at the Canada Basin (panels (a),(c),(e) in Figs. 14, 15), all model simulations predict a positive (too salty) salinity bias of $\approx 1$ psu in the upper $25-50$ m with respect to the ITP data, while exhibiting structures as similar as the ones for the Arctic Basin below the surface. E3SM (and partially RASM) also misses the subsurface temperature maximum at 50 m that is seen in the observations and WOA climatology, and which is associated with Pacific Summer Water. This bias is consistent with the reduced model heat flux through Bering Strait discussed

in section 4 (Fig. 8). One final point to note is that, while on average over the whole Arctic E3SM-Arctic-OSI has not drifted substantially from the first to the third cycle, a regional drift can be seen in the Canada Basin, more distinctively in the upper 100 m salinity (Figs. 14f, 15f) but partially in the temperature profile as well (Figs. 14a, 15a).

As described in the introduction, one science question that we hope to explore with E3SM-Arctic configurations revolves around the Arctic freshwater budget and freshwater content variability, and how it is tied to convective activities in the subpolar

North Atlantic. We therefore compute the simulated freshwater content (FWC) with respect to the typically used reference salinity of 34.8 psu. In Figure 16b,c we compare maps of Arctic FWC climatology computed over two time periods of the E3SM-Arctic-OSI simulation, specifically the first 15 years (contour lines in panel (b)) and the last 12 years of the simulation (shading in panel (c)), with an observational climatology computed from the Woods Hole Oceanographic Institution Beaufort Gyre FWC data over years 2003-2018[3] (shading in panel (b)). The climatology over the second period is also shown for E3SM-

LR-OSI (Fig. 16d) whereas the FWC over years 2005-2016 for the RASM simulation is presented in Fig. 16e. While the spatial pattern as well as overall magnitude of the E3SM-Arctic-OSI FWC is consistent with observations and RASM results in the early part of the simulation, the area of highest FWC within the Beaufort Gyre undergoes a positive trend during the second and third JRA cycle. This is mostly due to a drift in Arctic salinity, as can be seen from the increased depth of the reference salinity during the third cycle compared to the first (figures not shown) and from the full time series of Arctic surface-to-bottom

---

[2] Available for download at https://www.whoi.edu/page.do?pid=23096
[3] Available for download at https://www.whoi.edu/page.do?pid=161756

integrated salinity (not shown, but the salinity trend decreases from 0.13 psu per 100 years in the first JRA cycle, to 0.03 psu per 100 years in the third cycle). This salinity trend, much reduced in the third cycle compared to the second, is a model adjustment issue and is consistent with the global trend highlighted in Fig. 2c. Results for E3SM-LR-OSI are similar, although the high FWC pattern occupies even a larger area of the central Arctic compared with the E3SM-Arctic-OSI results.

Besides the FWC spatial pattern, an important feature for the model to reproduce is FWC variability. We therefore compute a Beaufort Gyre integrated time series of FWC anomaly, after removing the linear trend over the full duration of the E3SM-Arctic-OSI simulation and performing a 1-year running average (Fig. 16a). An anomaly computed from the same observational data used in Fig. 16a (Proshutinsky et al., 2019) is also plotted for comparison (green line; anomaly calculated by removing the overall annual mean). The model represents the general upward tendency of the observed Beaufort Gyre FWC over the last 16 years of the observed record, as well as the double peak in the FWC anomaly around the years 2010 and 2016. This is a very positive result and suggests that the processes responsible for this variability (e.g. ocean circulation and sea-ice variability and trend) are well represented in our current E3SM-Arctic-OSI configuration.

## 5.2 Sea-Ice Climatology and Trends

To analyze the state and evolution of Arctic sea ice simulated in E3SM-Arctic-OSI, we use the same approach for evaluating sea ice as the CORE-II project (e.g., Wang et al., 2016b) and the Coupled Model Intercomparison Project (CMIP; e.g., Stroeve et al., 2012). First, we examine time series of the Northern Hemisphere sea ice aggregated area in winter (February) and summer (September; Figure 17a). Results from the three JRA cycles are compared with the SSM/I derived observational estimates[4] (Cavalieri et al., 1999). The first JRA cycle shows an approximately 10-year long spin up adjustment from the initial sea ice state, which, as mentioned earlier, is a 1 m thick disk of sea ice extending poleward of $60°$. Overall, both February and September sea ice extent variabilities are represented very well (relative to the SSM/I data, their respective correlation coefficients (c) are 0.97 and 0.96); however, the absolute area from the model is overestimated by up to $1 \times 10^6$ km$^2$ compared with the observational estimates. Such differences are well within standard deviations of the CMIP6 multi-model sea ice area spread of $\pm 0.95 \times 10^6$ km$^2$ in March and $\pm 1.83 \times 10^6$ km$^2$ in September (SIMIP Community, 2020). It is also worth noting that the observational uncertainty, or the spread of observational sea ice area estimates is $\pm 0.54 \times 10^6$ km$^2$ in March and $\pm 0.66 \times 10^6$ km$^2$ in September. The actual magnitude of the bias is likely due to the cold SST biases, especially in the winter time Labrador Sea (compare Fig. 3, Fig. 14a,c and Fig. 15a,c). It could also be at least in part related to the fact that for the model area we are including areas of very thin ice ($< 20$ cm), which may not be detectable from (or transparent to) and accounted by satellites (W. Meier, National Snow & Ice Data Center (NSIDC), personal communication). The respective model and satellite trends, computed over years 1979-2016 of the third JRA cycle and over years 1979-2017, are $-3431$ km$^2$ year$^{-1}$ and $-3853$ km$^2$ year$^{-1}$ in February, and $-6526$ km$^2$ year$^{-1}$ and $-8219$ km$^2$ year$^{-1}$ in September. The simulated sea ice area also shows the acceleration of these trends that is apparent from observations since the late 1990's (Watts et al., 2021).

---

[4]Available for download at https://earth.gsfc.nasa.gov/cryo/data/arcticantarctic-sea-ice-time-series

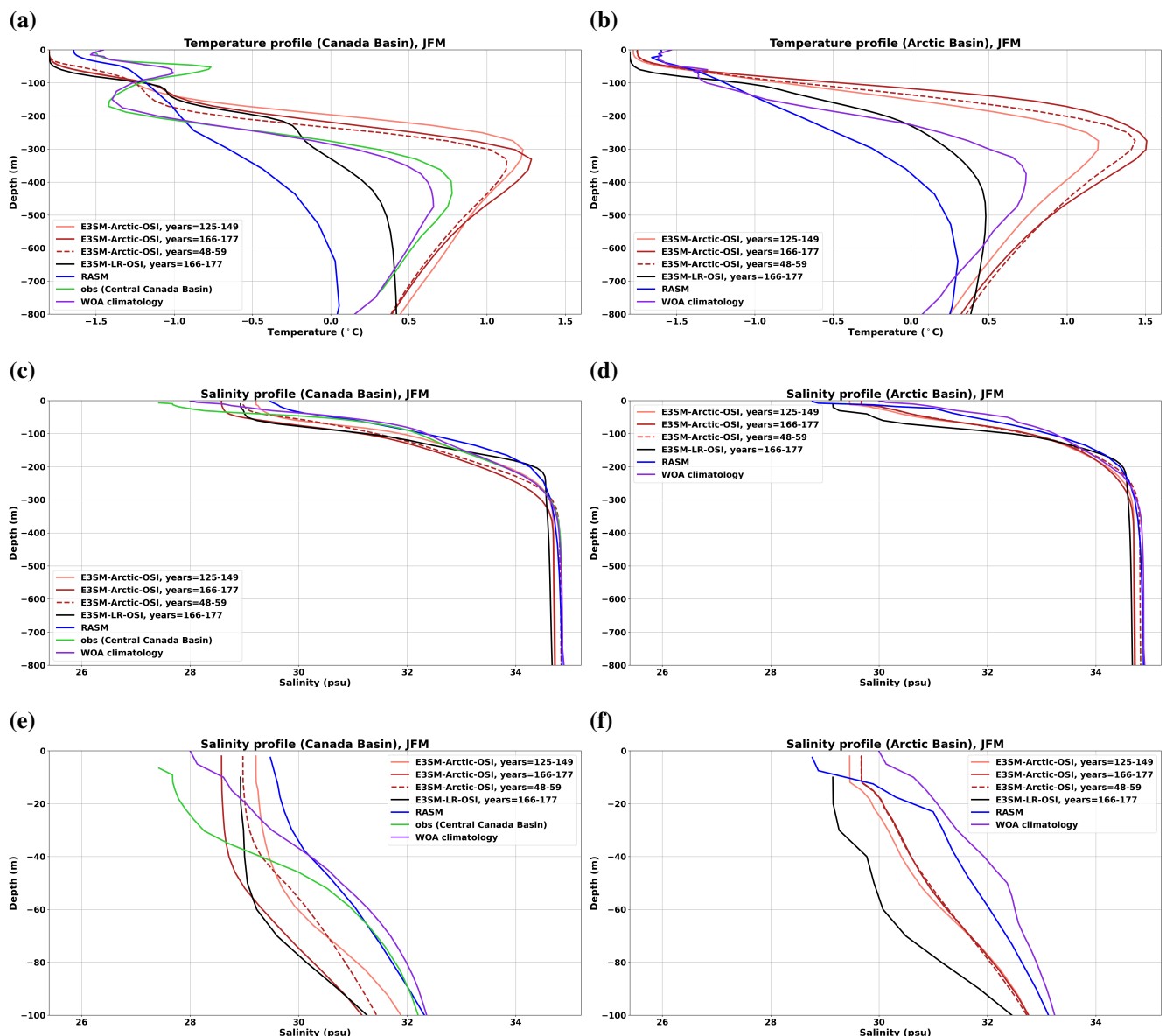

**Figure 14.** Vertical profiles of (a,b) temperature, (c,d) salinity, and (e,f) upper 100 m salinity for the JFM seasonal climatology, computed over the Canada Basin (a,c,e) and over the whole Arctic Basin (b,d,f; Barents and Kara Seas are not included in this calculation). Light red lines indicate E3SM-Arctic-OSI model climatologies computed over the early period years 125-149 of the third JRA cycle (corresponding to 1964-1988), whereas dark red solid lines are for model climatologies computed over years 166-177 (corresponding to 2005-2016, which is the period for which observations are available in the Canada Basin). Dark red dashed lines are for model climatologies computed over years 48-59 (end of first JRA cycle). Black lines represent E3SM-LR-OSI climatologies over years 166-177; blue lines indicate RASM model climatologies computed over years 2005-2014; purple lines indicate values from the WOA18 climatology; and finally green lines in (a,c,e) indicate observational values from the ITP buoys data in the Central Canada Basin.

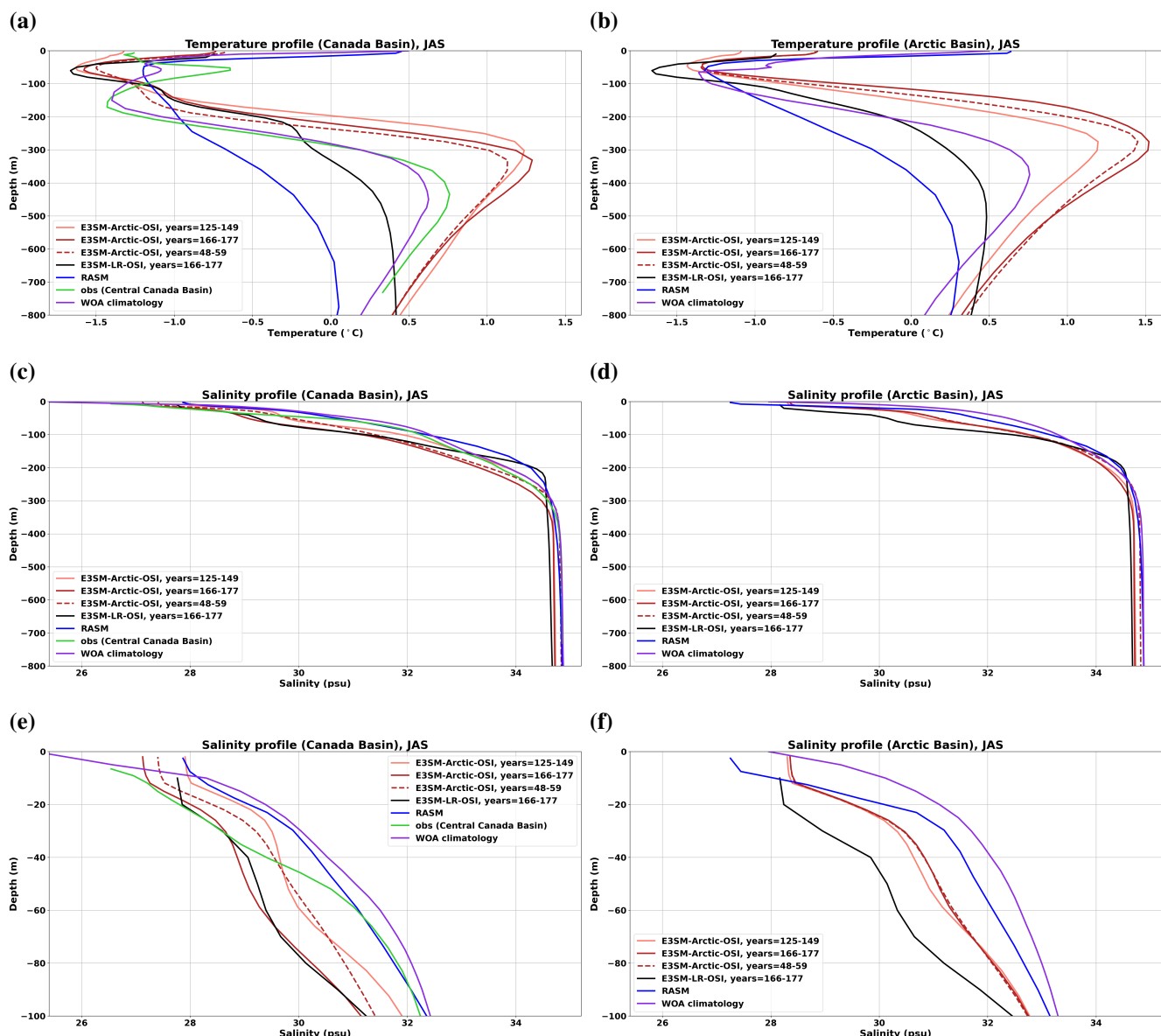

**Figure 15.** Same as Fig. 14, but for the JAS seasonal climatology.

The Northern Hemisphere aggregated sea ice volume for February and September is shown in Figure 17b. However, observational estimates are only available since 2003 from IceSat[5] (Yi and Zwally, 2009, updated 2014-04-15) and CryoSat-2[6] (Kurtz and Harbeck, 2017), only for parts of the year and with substantial uncertainties in estimated sea ice thickness and thus volume (Zygmuntowska et al., 2014; Bunzel et al., 2018; Tilling et al., 2018). Furthermore, model reanalysis estimates carry similar uncertainties in sea ice thickness and volume (Chevallier et al., 2017). Still, to provide some reference we use the Pan-Arctic Ice Ocean Modeling and Assimilation System (PIOMAS; Zhang and Rothrock, 2003) reanalysis sea ice volume estimates as an 'observational' proxy reference. The PIOMAS domain-wide sea ice volume bias has been estimated at $-2.8 \times 10^3$ km$^3$ for March and $-1.5 \times 10^3$ km$^3$ for October, with the corresponding trend uncertainty of 100 km$^3$ year$^{-1}$ over 1979-2010 (Schweiger et al., 2011). Even with the above ambiguity, it is clear that both the February and September time series of modeled sea ice volume are biased high, on the order of $5 - 7 \times 10^3$ km$^3$ in the 1980's and $3 - 5 \times 10^3$ km$^3$ in the 2000's. The respective E3SM-Arctic-OSI and PIOMAS trends are 39.0 km$^3$ year$^{-1}$ and 27.3 km$^3$ year$^{-1}$ in February, and 38.4 km$^3$ year$^{-1}$ and 31.6 km$^3$ year$^{-1}$ in September, which are comparable given the PIOMAS trend uncertainty. In addition, the respective model and PIOMAS time series of sea ice volume are significantly correlated in both February ($c = 0.86$) and September ($c = 0.80$).

The high bias in volume is further demonstrated in Figure 18, where E3SM-Arctic-OSI sea ice thickness distribution in March and September of 1980 and 2016 is shown. Since observations are only partially available, we compare the E3SM results to those of RASM by plotting RASM sea ice thickness for the same months mentioned above in Figure 19. E3SM-Arctic-OSI predicts excessively thick ice (greater than 4 m) over most of the western Arctic Ocean in 1980 as well as in winter 2016. We have also computed ice thickness differences between model climatologies and a mean distribution calculated from the 2003-2009 IceSat data record (not shown), and those results confirm that the overall model sea ice is $1 - 2$ m too thick in the western Arctic (with maximum bias in the center of the Beaufort Gyre), while being $1 - 1.5$ m too thin in parts of the eastern Arctic, especially during October-November. The ice thickness bias in E3SM-LR-OSI (not shown) is very similar in magnitude and structure to that of E3SM-Arctic-OSI, suggesting that horizontal resolution and internal variability in the sea-ice and ocean models are not affecting the simulation of sea-ice evolution, but rather that external atmospheric forcing may be a factor.

Despite the presence of these ice thickness biases, we consider very encouraging the fact that without any tuning the model produces a reasonable representation of the mean sea ice state, its seasonal to decadal variability, and trends. Based on previous studies and recent sensitivity experiments with the E3SM model, we believe that thick sea ice and the resulting high ice volume bias might be significantly reduced through optimization of model scale-aware parameter space.

## 6 Discussion and conclusions

Models with a regional grid refinement and a coarse resolution over the rest of the globe are expected to more realistically simulate the physics of refined regions at a fraction of the cost of an equivalent global high-resolution configuration. For such

---

[5]Available for download at https://nsidc.org/data/NSIDC-0393
[6]Available for download at https://nsidc.org/data/RDEFT4

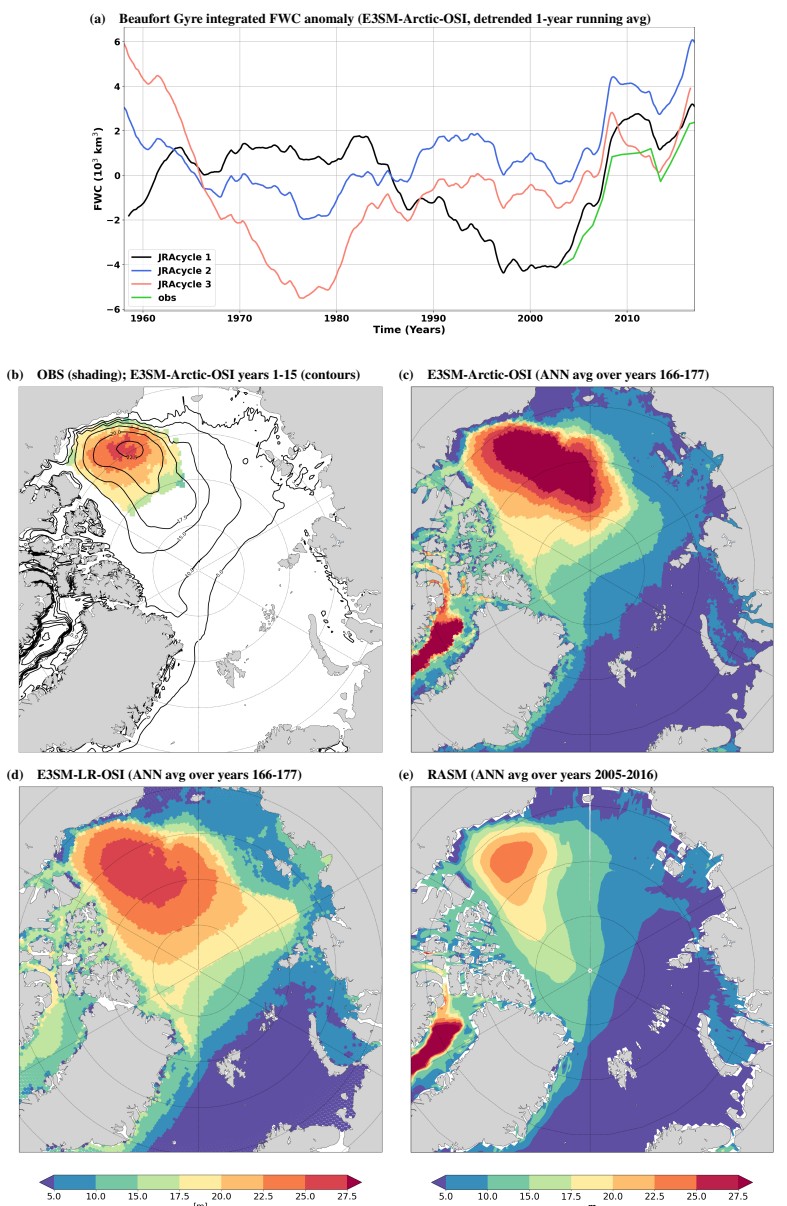

**Figure 16.** (a) Time series of Beaufort Gyre integrated freshwater content (detrended and 1-year running averaged) from the E3SM-Arctic-OSI simulation (black, blue, and pink lines indicate results from the three JRA cycles). Green line is the WHOI observational counterpart (anomaly computed by removing the 2003-2018 mean). (b-e) Annual climatology of freshwater content with respect to a salinity reference of 34.8 psu, computed from (b) observations over years 2003-2018, (c) E3SM-Arctic-OSI over the final 12 years of the third cycle (years 166-177, corresponding to 2005-2016), (d) E3SM-LR-OSI over years 166-177, and (e) RASM over years 2005-2016. Contour lines in panel (b) show the E3SM-Arctic-OSI annual climatology for the first 15 years of the first JRA cycle.

**(a)**

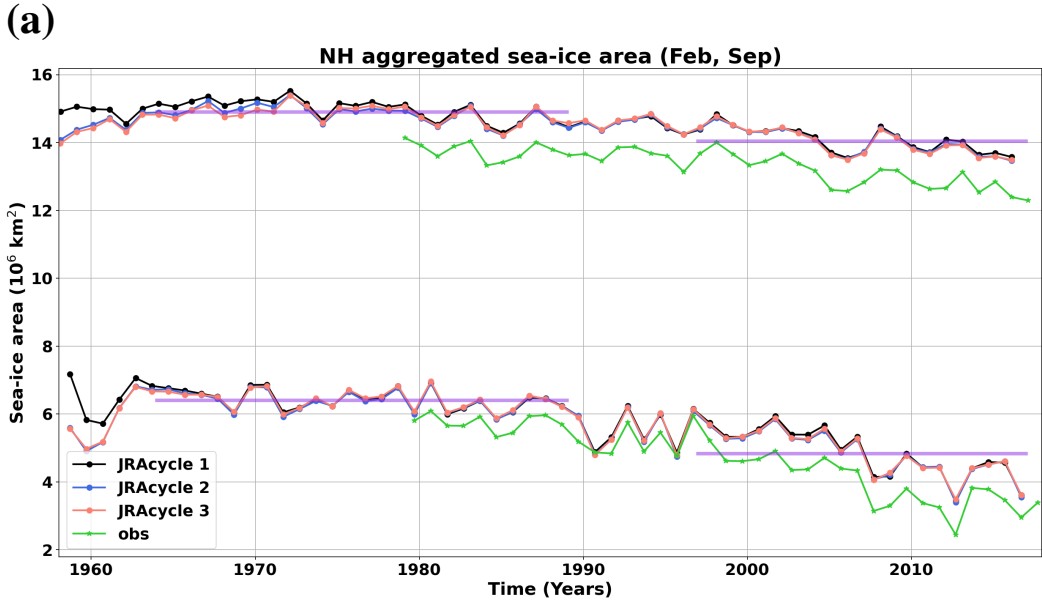

**(b)**

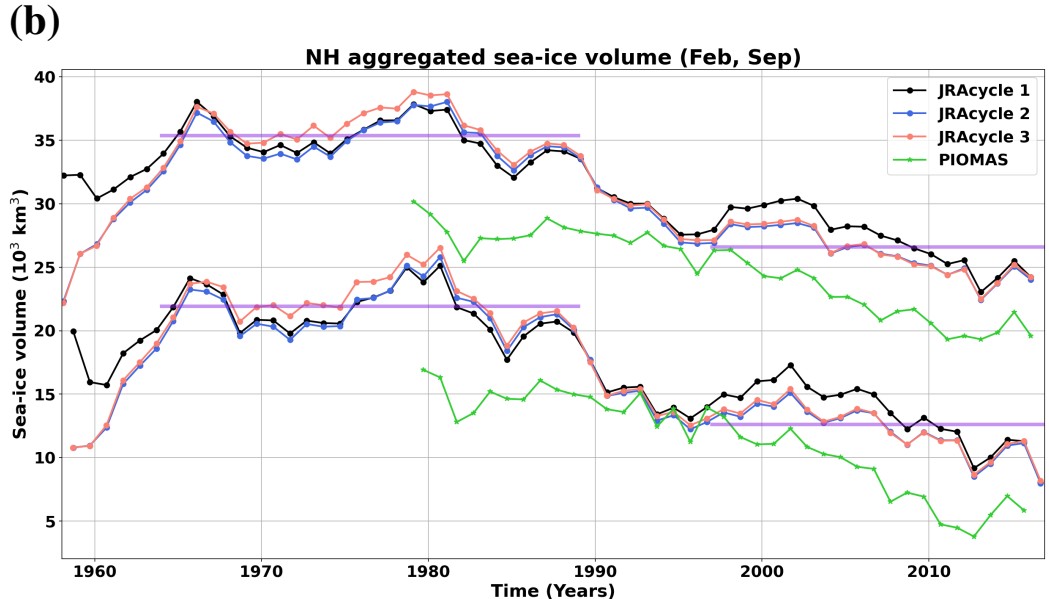

**Figure 17.** E3SM-Arctic-OSI time series of northern hemisphere aggregated (a) sea-ice area and (b) sea-ice volume, for the months of February and September. Results from the three JRA cycles are plotted as black, blue, and pink lines, using actual years instead of model years (similarly to Fig. 16a). Green lines represent SSM/I observations in the upper panel, and PIOMAS results in the lower panel. The two purple lines indicate the distinct averaged values of sea-ice area and volume for the first part of the cycles (years 1964-1988) and the second part of the cycles (years 1997-2016). Note that these two regimes correspond to the two periods used to compute separate climatologies in the vertical profiles of Figs. 14-15.

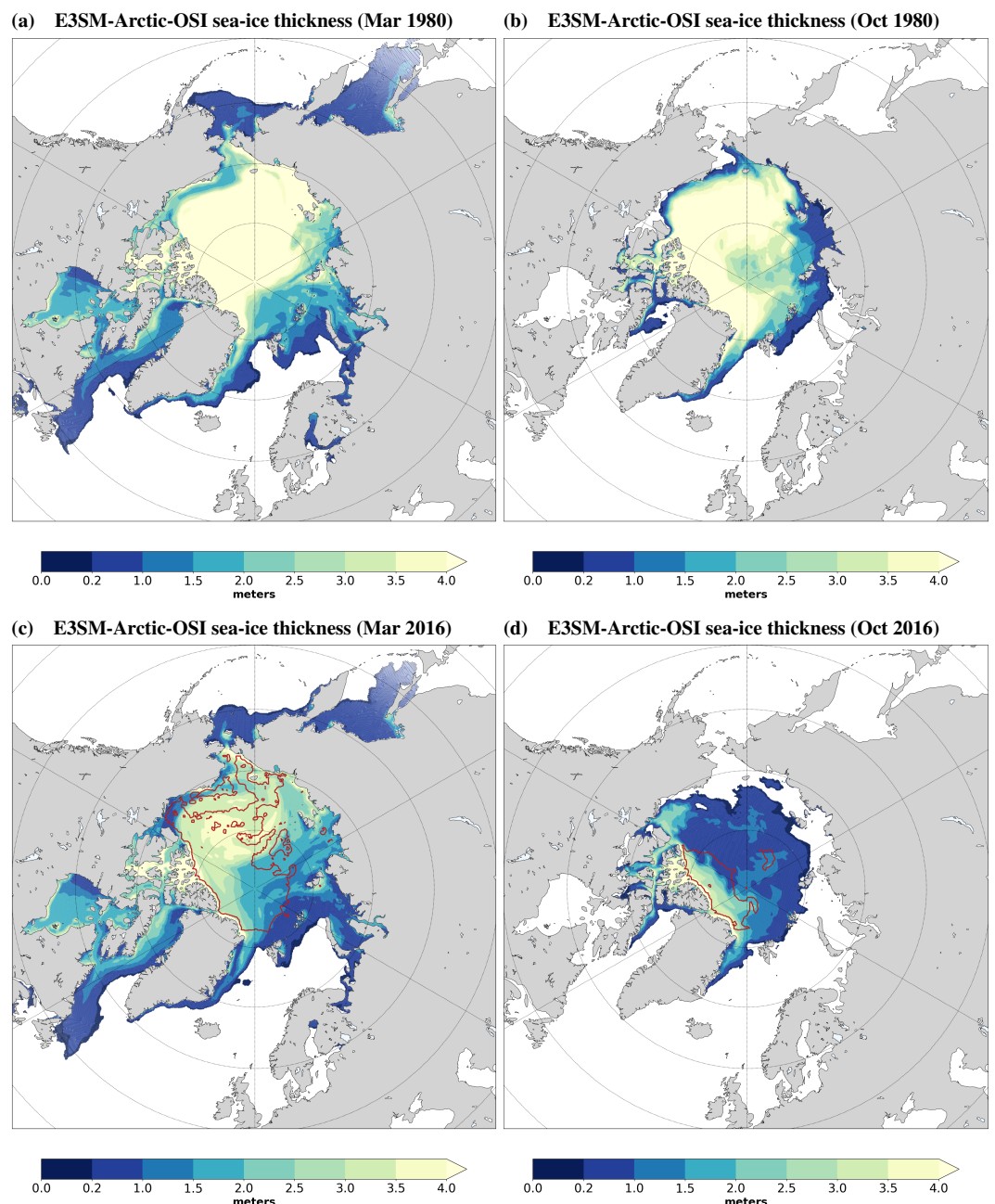

**Figure 18.** Maps of E3SM-Arctic-OSI sea-ice thickness for the months of (a) March and (b) October of year 1980 (third JRA cycle), and the months of (c) March and (d) October of year 2016 (again, from the third JRA cycle). Red contour in (c,d) indicates sea-ice thickness of 2 m from the CryoSat-2 satellite data (no satellite data is available in 1980, hence the missing red contours in (a,b)).

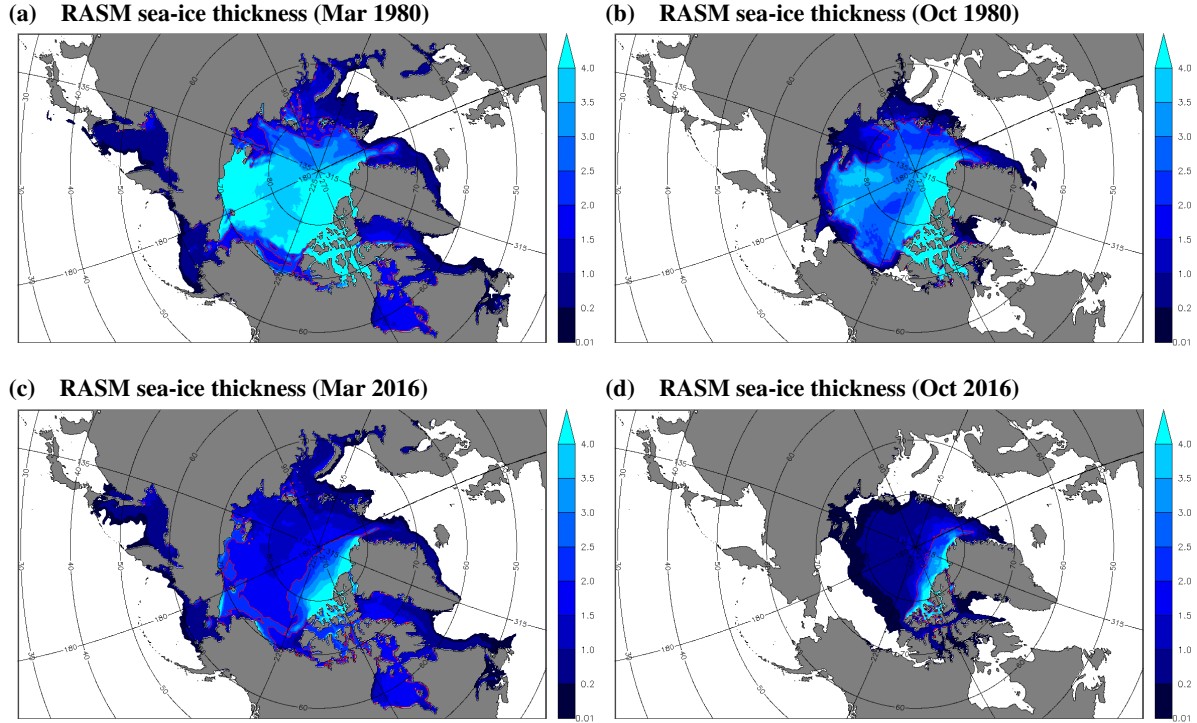

**(a) RASM sea-ice thickness (Mar 1980)**  **(b) RASM sea-ice thickness (Oct 1980)**

**(c) RASM sea-ice thickness (Mar 2016)**  **(d) RASM sea-ice thickness (Oct 2016)**

**Figure 19.** Similar to Fig. 18, but for the RASM results. Red contour in this case indicates 2 m ice thickness from the model results. Note that colorbar values are identical to those in Fig. 18.

models like E3SM that so far do not use adaptive time stepping, this obviously comes with the caveat that to ensure numerical stability, the grid cells with the highest horizontal resolution determine the global time step. Therefore, there will always be a limit on the horizontal resolution of refinement, beyond which configurations become no longer advantageous compared to
global high-resolution models. More importantly, two aspects of modeling the Earth System must be considered when designing regional refined meshes. Firstly, local physics may be influenced by remote dynamics, which should be taken into account when focusing resolution in certain places versus others. Secondly, some model biases are not readily solved by increasing resolution, but are rather due to inaccurate representation of physics, lack of observational baseline, inadequate parameterizations, or issues with coupling between different components of the ESM. Scale-aware parameterizations are another challenge
for unstructured and refined global grids. The type of biases described above are often model dependent and to be resolved, they require in-depth investigations, additional measurements, optimization of parameter space, different coupling strategies, and possibly additional model development.

In the E3SM-Arctic-OSI configuration described in this paper, we have forgone the inevitable complications associated with coupling the ocean and sea-ice model components to the actively evolving atmosphere and land components. Our motivation
was to first focus on the consequences of the pan-Arctic mesh refinement in the ocean and sea-ice components on their simu-

lation of that region. We have taken the approach, summarized in this paper, to evaluate the ability of the model to reproduce certain prioritized metrics that are deemed important to the realistic representation of the pan-Arctic climate dynamics and its connection to the lower latitudes. We have found that many metrics, such as the sea-ice climatologies, variabilities and trends, the variability of the freshwater content of the Arctic, the exchanges through the Arctic gateways and their vertical ocean prop-
410 erties are satisfactorily represented in E3SM-Arctic-OSI (where by 'satisfactorily represented' we intend with respect to the uncertainty and variability of the observations). However, still a few important aspects, such as the sea-ice thickness distribution and the upper 100 m Arctic ocean stratification, are misrepresented in similar ways across different E3SM configurations, and are therefore likely a result of model-specific biases and/or model tuning, or lack of it, in the ocean, sea-ice, and atmosphere. The sea-ice in particular is very sensitive to the simulated ocean state or prescribed conditions from the reanalysis atmospheric
forcing, in addition to choices of scale-aware sea-ice parameters. In our future E3SM-Arctic configurations, we will be adopting the E3SM v2 model version, which features improvements in the ocean eddy parameterization that should help reproduce the ocean stratification in places where mesoscale eddies are unresolved. We also plan to more systematically examine the sensitivity of sea-ice thickness to varying internal parameters and coupling with the ocean and atmosphere, to reduce biases in the simulated sea-ice thickness distribution.

Moreover, such metrics as the Arctic ocean stratification below 100 m and the AMOC strength, although not perfect, are improved in E3SM-Arctic-OSI compared to the global low resolution version of the model (E3SM-LR-OSI). These results suggest that such metrics could be additionally improved by refining resolution in the subpolar gyre and/or in the western boundary current region.

Finally, future E3SM-Arctic configurations will couple the ocean/sea-ice components with an active atmosphere (and land)
component. We will consider coupling the regionally refined ocean/sea-ice mesh first with a low-resolution, standard atmosphere mesh and later with an Arctic regionally refined configuration, which will be more appropriate for reproducing storms propagating from the North Atlantic and Pacific Oceans into the Arctic.

*Code and data availability.* The version of the E3SMv1 model used to run E3SM-Arctic-OSI and E3SM-LR-OSI is publicly available under Zenodo, doi:10.5281/zenodo.5548434. The model data used for the analysis presented in this paper is also available through Zenodo,
doi:10.5281/zenodo.5548528.

*Author contributions.* MV led the writing of the manuscript and the paper analysis with key contributions from WM, YL, and WW. The E3SM experiments were run by GD and RO. MP and JW helped with making the 60to10 MPAS mesh and with the setup of the E3SM-Arctic-OSI configuration. TC, DC, and AT helped during the running of the experiments. All co-authors commented on the final version of the paper.

*Competing interests.* No competing interests are present.

*Acknowledgements.* We thank Mathew Maltrud for help with configuring the model and Xylar Asay-Davis for help in finalizing MPAS analysis scripts. This research was supported by the Regional and Global Model Analysis component of the Earth and Environmental System Modeling program of the U.S. DOE's Office of Science, as contribution to the HiLAT-RASM project. MP, JW, and AT were supported by the E3SM project. The E3SM-Arctic-OSI simulation as well as most of the analysis of this paper were performed at the National Energy Research Scientific Computing Center (NERSC), a U.S. DOE's Office of Science User Facility operated under Contract No. DE-AC02-05CH11231. The E3SM-LR-OSI simulation was performed at the U.S. Department of Defense High Performance Computing Modernization Program (HPCMP). We also acknowledge the use of computer resources at the Institutional Computing facilities at LANL.

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
