# Peer review of "An evaluation of the E3SMv1-Arctic Ocean/Sea Ice Regionally Refined Model"

_Geoscientific Model Development, 2021_

## Author Response (AR1)

**Response to reviewers' comments for: An evaluation of the E3SMv1-Arctic Ocean/Sea Ice Regionally Refined Model**

January 28, 2022

**Response to Reviewer 1**

Dear Dave, thank you so much for your careful and thorough review of our manuscript. You have made very valid points, which we believe we have addressed by modifying/adding to the manuscript text and responding to your comments below. Please note that your original comments are in italics. Line numbers, as well as Figure and Table numbers, indicated in our responses refer to the new version of the manuscript.

**Major comments**

1. *My first major comment is that this is kind of a rehashing of Petersen et al. 2019. While these simulations are forced with JRA as opposed to CORE, the model details are the same in both papers as I see it. The regionally refined aspect is new, but it would be helpful for the authors to contrast the model details from this paper and Petersen et al. 2019. Perhaps a table would be helpful here.*

   Good point. We have now added Table 1, comparing key features of the E3SM-Arctic and E3SM-LR configurations described in the manuscript against those documented in three previous E3SM publications: Petersen et al. 2019 (CORE-forced E3SM-HR and E3SM-LR simulations), Golaz et al. 2019 (fully coupled E3SM-LR simulations), and Caldwell et al. 2019 (fully coupled E3SM-HR simulations).

2. *One of the challenges of regionally refined grids is the scale aware parameterizations. I would like to see more details here about how the parameterizations adapt from the coarse resolution part of the domain to the finer resolution part of the domain. Again, this would be an interesting thing to contrast with Petersen et al. 2019.*

   As described in the manuscript (lines 90-99), the GM kappa transitions from 0 to 600 when the horizontal resolution goes from 20 to 30 km. Considering how the average resolution varies meridionally (Fig. 1a), this means that: i) the transition region falls between the white and red horizontal lines in Fig. 1b; ii) kappa=0 for latitudes above the red line; and iii) k=600 for latitudes below the white line. We have now added a clarification sentence on lines 98-99, and also compared the GM treatment to that used in Peterson et al. 2019, Golaz et al. 2019, and Caldwell et al. 2019 in Table 1.

3. *I am curious why the authors chose 3 cycles of JRA? I see the salt and temperature trends are still very strong here and particularly above 1000 m depth. This seems to accelerate in the 3rd cycle. Are you using some kind of temperature or salinity restoring? This is often required in a data atmosphere forced mode. I see now where the restoring is 1 year. This is fairly weak. A discussion of how the restoring would impact the drift would be a good addition here.*

   Those are very good questions. As the reviewer is aware, there is always a trade off in running unconstrained (or partially constrained, as in this case) climate models, between reaching some semi-equilibrium (in the global T and S trends, for example), and trying to keep the drift to a minimum at the same time. We decided to put this trade off at the third JRA cycle, motivated by the fact that all the trends in top-to-bottom T, S, Arctic gateway transports, and aggregated Arctic

sea-ice area and volume are quite similar between the second and third cycle. As for the OHC trend (Fig. 2a), we agree that the upper 700 m heat content is still trending up during the third cycle, but the top-to-bottom trend has certainly reduced from the second to the third cycle. Finally, the choice was also a practical one, as computational resources available to us were scarce at the time when these simulations were made.

We also agree that the 1-year SSS restoring time scale for simulations that are forced by JRA55 may be too weak. We did re-run one cycle with a 1-month restoring scale, and we found some slight improvements in the upper ocean salinity and the mean AMOC (approximately 1 Sv improvement in the maximum AMOC at 26.5° N). We are certainly considering changing the default SSS restoring time scale to something smaller than 1-year for our future JRA-forced runs.

4. *With the drift, it is sort of difficult to compare the third cycle of the E3SM simulations to observations or only to one cycle of RASM. Perhaps it would be better to compare the first cycles.*

As mentioned in the response to the previous comment, the trends in some key fields of interest to us did improve in the third cycle compared to the first cycle, and for this reason we prefer to show the results mostly from the third cycle. As suggested by the reviewer, we did include Arctic stratification information from the first cycle in Figures 14, 15 (see response to reviewer's comment later in this document).

5. *It is interesting that the MOC seems to be fairly well equilibrated (Figure 5b) after 1 cycle despite the heat and salt trends. Also, the MOC max transport is significantly higher in the E3SM-Arctic simulation. It would be interesting to contrast with Bryan, Gent, and Tomas (2014) here in which they found that differences in parameterizations at one degree and 0.1-degree led to differences in the poleward heat transport and hence sea ice differences.*

Good points. It should be noted that, contrary to the configurations analyzed in Bryan et al. (2014), both the E3SM-Arctic-OSI and E3SM-LR-OSI configurations considered in our study feature the same low resolution (and GM treatment) in the Southern Ocean. We do believe though that the lack of GM parameterization in the Arctic and subarctic in E3SM-Arctic allows us to better resolve the stratification in those regions of the subpolar gyre where North Atlantic water masses are formed, therefore yielding a stronger AMOC in the E3SM-Arctic simulation. We have now added a sentence discussing these points on lines 173-176.

6. *Figures 7, 8, and 9 are interesting. The Nares strait is not resolved in the LR simulations as mentioned. I would like to see more discussion on how this impacts the downstream transport and the MOC. Is there a shift in transport from the Canadian Arctic to Fram Strait?*

Partly. Considering the mean averages of volume transport across the different gateways from Table 2 (and Fig. 7), we have the following differences between E3SM-LR-OSI and E3SM-Arctic-OSI: $\approx 0.4$ Sv out of Davis Strait instead of 1.6 Sv (deficit due to the absence of flow out of Nares Strait in the low-res configuration); $\approx 2.5$ Sv out of Fram Strait instead of 2.2 Sv; and $\approx 2.9$ Sv in from the BSO and Bering Strait instead of 3.8 Sv. Therefore, the reduced outflow out of Davis Strait can be explained by a slightly higher outflow out of Fram Strait (as the reviewer mentioned), but also by a fairly reduced net inflow to the Arctic from both the Barents Sea Opening and Bering Strait. We have now added a short discussion about this on lines 238-240.

7. *Figures 10, 11, and 12 are helpful in understanding the vertical distribution differences. How much do these change between cycles? That is, how are they impacted by the drift.*

We can compare panels a,c,e of Figures 10-12 to the corresponding climatologies computed over years 40-59 (last 20 years of the first JRA55 cycle), which are presented in Figures 1, 2 below. We can see that some changes in temperature and salinity are apparent below the Atlantic Water layer in Fram Strait and the BSO, and in the West Greenland Current in Davis Strait, but the overall structure of the gateways stratification is quite consistent between the first and third JRA cycle, and consequently the velocity structures are also very comparable.

8. *Figure 14 and 15 show the central Arctic profiles of temperature and salinity. This is one of the areas where I am worried about the comparison of the third cycle in the E3SM runs to the first cycle in the RASM runs. The drift in the model causes these profiles to move further from the initialization*

[Figure]

Figure 1: Cross-section of (a,b) potential temperature, (c,d) salinity, and (e,f) normal velocity for Fram Strait and the BSO for E3SM-Arctic-OSI. Annual climatologies are computed over years 40-59. Black contours show potential density (sigma0).

[Figure]

Figure 2: Similar to Fig. 1 but for Davis and Bering Straits.

*from PHC. This is always an issue with ocean models. That said, are the RASM runs initialized differently? What is going on here with the RASM temperature profiles? Is vertical resolution playing a role here?*

Regarding the model drift and its effect on the results presented in Figures 14, 15: we have now included the profiles for the last 12 years of the first JRA cycle, in addition to the ones computed over the two periods of the third JRA cycle. They give a good idea of how the drift affects the stratification over the whole Arctic and locally over the Canada Basin only.

As for the questions regarding the RASM results: yes, the RASM run was initialized differently, using ice-ocean restarts after a 75-year long spin up forced with CORE2-CIAF reanalysis, which we have diagnosed as the primary cause of the temperature bias. More recent RASM simulations initialized from a new spin up forced with JRA55 do not experience such a cold bias in the Atlantic Layer. More details about RASM and the simulation used here are now included on lines 118-133.

9. *In addition to the above, some of the most interesting seasonal differences in 14 and 15 are above 100 m. Instead of density, perhaps you could zoom in on the top 100m instead. The surface temperature and salinity biases are indicative of some ice-ocean coupling issues. Do the salt and fresh water fluxes assume 4 psu in the sea ice?*

We agree with the figure revisions suggested by the reviewer. The density panels of Figs. 14, 15 have now been substituted with the upper 100 m close up of salinity. Text has been adjusted accordingly (lines 292-293).

Yes, the salt/fresh water fluxes assume that sea ice has salinity of 4 psu.

10. *Figure 16 is interesting, but might be a candidate for removal. Perhaps a discussion and leading to another paper. I feel like there could be so much more here and the discussion is fairly limited.*

Considering the science questions that we would like to investigate in future studies using E3SM-Arctic, we feel strongly that we should document the freshwater content for this simulation. Therefore, we would like to retain Figure 16.

11. *It is nice to see that the sea ice volume and area are consistent across the three cycles for the most part. They impacted a bit by the upper ocean drift. There is something funny in the volume before 1995 and after. The trend is faster than PIOMAS before 1995 and then slower after 1995. I worry this is an artifact of the JRA forcing.*

Yes, we also suspect that the differences in sea ice volume trends compared with those from PIOMAS might be related to the JRA55 forcing, especially considering that the RASM volume trends are similar to those from E3SM-Arctic-OSI. We are currently exploring options to correct this known atmospheric reanalysis Arctic warm bias in winter.

12. *On the JRA forcing. Are you doing corrections over the sea ice? There are definitely some biases in the JRA in the polar regions. While there were corrections added some biases remain and different than the CORE forcing.*

No, we are not applying any correction to the JRA forcing. It may also be worthwhile to mention here that we are using version v1.3 of the JRA55-do data set for this set of simulations.

13. *What are the lateral boundary conditions for the sea ice and ocean in the RASM runs?*

This information has now been included in the more detailed description of RASM on lines 118-133.

14. *In Figure 19, I find it interesting that all of the model simulations have a thickness bias in the Beaufort Sea. It might not be fair to compare longer periods in the model to such a short period of IceSat. There could be some substantial interannual variability here. The biases near the CAA might just be something in the IceSat data. I'm not sure this figure is helpful. Also, I wonder if the "bias" in the Beaufort Sea thickness could be related to freshwater content in the Beaufort gyre? Maybe adding the RASM thickness in Figure 18 instead?*

We agree with the reviewer points here and have now included a new Figure 19 showing RASM results corresponding to the same months and years shown in Figure 18 from E3SM-Arctic-OSI. We have adjusted the discussion in the text accordingly on lines 364-373.

15. *Back to my earlier point about scale aware parameterizations. Does this feed into the discussion and conclusions about the regionally refined configuration versus the low res configuration? Also, the JRA biases are key here. Ultimately, even though one improves resolution in the sea ice, the thickness distribution in particular is generally set by the wind patterns as mentioned earlier in the manuscript.*

We agree with the points made here. We have added a sentence about the challenges of scale aware parameterizations in the associated discussion in Section 6 (lines 388-389).

**Minor comments**

1. *On line 65, please delete "occasionally".*

Done.

2. *How are the sea ice albedo parameters set/tuned?*

For this version of the model and particular set of simulations, we have not done tuning experiments with sea-ice parameters (we are doing them in the more recent E3SM-Arctic simulations that we have been running). Therefore, albedo/shortwave parameters are set as in CESM. Here is a list of them:

```
config_shortwave_type = 'dEdd'
config_visible_ice_albedo = 0.78
config_visible_snow_albedo = 0.98
config_infrared_ice_albedo = 0.36
config_infrared_snow_albedo = 0.70
config_max_melting_snow_grain_radius = 1500.0
config_temp_change_snow_grain_radius_change = 1.5
config_variable_albedo_thickness_limit = 0.3
```

3. *I see this is the B-grid versions of the models. Any thoughts about C-grid? How does the sea ice compare to Turner et al. (2021) which is in review?*

MPAS-Ocean uses a C-grid, whereas MPAS-Seaice uses the B-grid. The version of MPAS-Seaice is the one described in Turner et al. (2021), which is also the one available in E3SM. There is current work in progress on moving MPAS-Seaice on the C-grid.

4. *Some more details of the RASM simulation would be helpful.*

Thanks for the suggestion. A more detailed description of RASM has now been included on lines 118-133.

**Response to reviewers' comments for: An evaluation of the E3SMv1-Arctic Ocean/Sea Ice Regionally Refined Model**

January 28, 2022

**Response to Reviewer 2**

We thank the reviewer for the helpful comments and the positive review of our manuscript. We believe we have addressed the reviewer's concerns and suggestions in our response below. Please note that the reviewer's original comments are in italics. Line numbers, as well as Figure and Table numbers, indicated in our responses refer to the new version of the manuscript.

**Comments**

1. *line 4: ...cost of high – resolution (HR)* **regular gridded** *global configurations...*

   We have clarified this sentence (lines 4-5) as follows: "...relative to the respective cost of configurations with high-resolution (HR) everywhere on the globe.".

2. *line 6: "...while employing data-based atmosphere, land and hydrology components..." If I understand well you run MPAS+Sea-Ice more or less in a standalone mode with a prescribed atmosphere. What are in this case the land and hydrology components? Maybe some rephrasing is necessary.*

   We do not run MPAS in standalone mode, but as the ocean and sea-ice components of E3SM. We run E3SM in data-atmosphere/data-hydrology and active ocean/sea-ice mode (so called G-case in E3SM/CESM lingo). The data used to forced MPAS is described in Section 2 of the manuscript: it's basically the JRA55 atmospheric data, which also supplies the river runoff data (only land-related feature that is needed for this ocean/sea-ice configuration).

3. *line 43: "...Wang et. al 2018...", Its fully OK here to cite Q. Wangs paper, but since you anyway run your configuration uncoupled it might be worth it to cite also the papers of C. Wekerle et. al 2013, 2017a and 2017b. Since they directly deal with mesh improvements in the Arctic realm (but in an uncoupled FESOM standalone configuration) and their consequences for the local ocean circulation down to an eddy resolving regime (C. Wekerle et. al 2017b).*

   We agree that we should have added the reference to the Wekerle et al. publications, and have done so on line 42.

4. *line 75: Why only 10 km was chosen for the Arctic refinement, considering the rossby radius for high latitudes, at this resolution the Arctic ocean will be barely eddy permitting. For example the standard higher resolved Arctic FESOM configuration goes down to a resolution of 4.5km using around 650k surface vertices. There exists an intentional similar paper to yours within the FESOM community (C. Wekerle et. al 2017a), maybe it's worth to be cited.*

   We agree completely with the reviewer statement about the need for higher resolution in the Arctic, in order to better resolve the local Rossby radius of deformation. We did run a simulation similar to the one described in the manuscript but using an Arctic regional refinement of 6 km instead of 10 km. Unfortunately that simulation, while being approximately 3 times more expensive than the 10-km refined mesh, did not produce any significant improvements in the Arctic and subarctic ocean and sea-ice representation. We concluded that a resolution of at least 3-4 km is necessary to really

make a difference in the Arctic (and likely in the Southern Ocean as well). This very-high-resolution Arctic configuration of E3SM is in the planning stages at the moment, and we hope to have some results available over the next few years.

5. *line 88-89: Why is there no background diffusivity utilized?*

We have tried using some background diffusivity in recent E3SM-Arctic runs, but the constant value of diffusivity everywhere does not work well for regionally refined meshes, because the model becomes unstable in the shallow regions with highest horizontal resolution. We have also tried background diffusivity in low resolution E3SM runs, but little change has been observed in the simulated T, S, and AMOC strength. We have concluded that more natural ways of representing deep ocean mixing, such as tidal mixing, should be instead used in MPAS-Ocean. We are working on having tidal mixing available in the next version of the model.

6. *line 110: I would like to know but also the community might like to know at which time-step the high and low resolution configuration perform.*

We have now added the requested information (10 minutes baroclinic time step for the high resolution configuration and 30 minutes for the low resolution configuration) on lines 90 and 111.

7. *line 111-118: I haven't fully got the point why or for which purpose you bring the RAMS configuration into the comparison since it's also just another model that is not directly related to E3SM. Maybe you can clarify in a couple of sentences, also in the introduction, what's the benefit of RAMS in this comparison.*

We thank the reviewer for suggesting this. We have now added the following sentence in the introduction (lines 56-59): "We also compare results with a high-resolution Regional Arctic System Model (RASM) simulation when observations are scarce or unavailable. Due to the higher number of constraints and its Arctic focus, we expect RASM to give a realistic representation of local processes, while obviously not directly accounting for the Arctic to mid-latitude exchange processes".

8. *line 132-136: For my own curiosity (doesn't need to be in the paper), can you say something to the "...recent improvements in the MPAS-Ocean eddy parameterisation scheme...".*

Absolutely. The next release of E3SM (v2 version) will include a version of MPAS-Ocean with the following options for the mesoscale parameterization scheme: i) inclusion of Redi mixing, with Redi kappa constant as well as variable in space and time; ii) variable GM kappa (again in space and time) following a number of different algorithms; iii) the ability to transition from GM-on to GM-off based on the local Rossby radius of deformation and local horizontal resolution.

9. *line 145: Are there known causes why your AMOC in the high but especially in the low resolution are so weak?*

Good question! We currently have a E3SM special working group that is trying to understand and untangle this very problem. The conclusion so far is that the ocean model is not fully able to mix or distribute the fresh water inputs/caps that form in the upper polar oceans at low resolution. Since the high-resolution simulation (Caldwell et al. 2019) displays a strong AMOC, we believe that vertical mixing is not the major culprit, but rather the implementation of the mesoscale parameterization must be at least partially responsible for the upper ocean fresh biases that we see in many E3SM low resolution simulations. We are hopeful that these investigations will be part of an upcoming publication.

10. *line 145-148: I would be interested to know what the maximum March (NH) and September (SH) mixed layer depth in both configurations looks like.*

We have not stored maximum mixed layer depth (MLD) for these particular simulations, but we have plotted the average MLD for March (NH) and September (SH) in Figure 1 below, as requested by the reviewer. It is interesting to note that some differences are even found in the Southern Hemisphere (where both configurations feature the same horizontal resolution), but the main differences are found in the Northern Hemisphere, with higher MLD found in the Greenland, Irminger, and Labrador Seas in E3SM-Arctic-OSI, as opposed to the Norwegian Sea in E3SM-LR-OSI.

[Figure]

Figure 1: Average mixed layer depth for March (upper panels, Northern Hemisphere) and September (lower panels, Southern Hemisphere) for E3SM-Arctic-OSI (left panels) and E3SM-LR-OSI (right panels). Monthly climatologies are computed over years 40-59.

11. *line 211: ...heat flux through Davis Strait...*

    Fixed.

12. *line 320, Fig.14 and Fig.15: You compare the 3rd. cycle of your E3SM simulations with the 1st. cycle of the RAMS simulation. In my experience, there are usually considerable differences between the 1st. and the 3rd. forcing cycle. At least this is the case for the Atlantic and Southern Ocean. For the Arctic these differences might be not that large, but nevertheless it might be of benefit also to provide the temperature and salinity profiles of the 1st. forcing cycle of your E3SM simulation in Fig. 14 and Fig. 15 as a dashed line.*

    As suggested by the reviewer, we have now included the profiles for the last 12 years of the first JRA cycle as dark red dashed lines in Figures 14, 15, in addition to the profiles computed over the two periods of the third JRA cycle. They give a good idea of how the stratification changes from cycle to cycle over the whole Arctic and locally over the Canada Basin only.

---

## Editor Decision (ED1)

An evaluation of the E3SMv1-Arctic Ocean/Sea Ice Regionally Refined Model,
by Veneziani et al

Dear Author,

Thank you for your revised manuscript. Before considering your paper for publication I would like you to address the following remarks. In general, I think you answered the reviewers' remarks satisfactorily but that you should add some text in the manuscript about the issues raised (and not only reply to the reviewer).

Reviewer #1

- Major comment #1
  - Thank you for adding Table 1. In the legend, can you be more specific about the two columns for « This paper » (I suppose the left one is for E3SM-Arctic-OSI and the right one for E3SM-LR-OSI) and about the two colums for Petersen et al. 2019 (I don't know what the two columns refer to).
  - Why isn't the RASM simulation described in this table ?
  - Also in the text where you introduce Table 1, L112-113, please give a short written description of the differences so to help the reader understanding Table 1.
- Major comment #2 : Please provide some details about how parameterizations other than GM vary with resolution ; if they don't please specify that.
- Major comment #3 :
  - Please add details in the text to justify the use of 3 cycles only.
  - Furthermore, for me a « trend » is a variation of a quantity per unit of time. I think you should change on L141 « a cooling trend by up to 0.5◦C «  by « a cooling persistent anomaly of up to 0.5◦C ».
  - Why is the  upper ocean freshening more concerning than the other, especially more than the OHC 0-700 one ?
- Major comment #4 : I understand that you better resolve the stratification in those regions of the subpolar gyre where North Atlantic water masses are formed because you do not activate GM in the Arctic and subarctic in E3SM-Arctic. If this is the case, I think L.175 «current GM implementation in MPAS-Ocean » is confusing. Maybe replace «current GM implementation in MPAS-Ocean » with « activation of GM in MPAS-Ocean ».
- Major comment #7 : Please add in the text some of your conclusions reported to the reviewer, even if you don't add Figures 1 and 2 from your reply.
- Major comment #8: Currently, you just added « In addition, E3SM- Arctic-OSI results are also shown for years 48-59 (end of first JRA55 cycle), in order to compare the stratification from the third and first cycles (solid and dashed dark red lines in Figs 14-15). » but you don't draw any conclusion. Please add in the text some of your conclusions reported to the reviewer, i.e. the fact that the drift affects the stratification over the whole Arctic and locally over the Canada Basin only ».

- Major comment #9: Is the fact that salt/fresh water fluxes assume that sea ice has salinity of 4 psu mentionned somewhere in the text ? If not do so, as the reviewer explicitely asked for that information.
- Major comment #11: Can you add something about this in the text ?
- Major comment #12: I don't see anywhere in the text that you version v1.3 of the JRA55-do data set, can you add this and also something about the fact that you don't do any correction over ice ?
- Minor comments #2 and #3 : Can you clarify these details somewhere in the text ?
- Minor comment #12: Thanks for adding the profiles for the last 12 years of the first JRA cycle as dark red dashed lines in Figures 14, 15. In your reply, you write « They give a good idea of how the stratification changes from cycle to cycle over the whole Arctic and locally over the Canada Basin only. » ; can you explicitly detail your observations on this in the text ?

Reviewer #2

- Comment #4 : These are interesting remarks ; please add something about this in the text.

My additonal comments :

- Table 2 : The captions mention  E3SM-Arctic-OSI, E3SM-LR-OSI and observations but not RASM, which is indeed included in the Table; please modify the captions.
- L 409 : remove « they » at the end of the line

---

## Author Response (AR2)

**Response to editor's comments for: An evaluation of the E3SMv1-Arctic Ocean/Sea Ice Regionally Refined Model**

February 24, 2022

We thank the editor for the suggestions and the careful review of the manuscript. We believe that we have addressed all the raised concerns and comments. Please note that the editor's comments are indicated in blue; line numbers, as well as Figure and Table numbers, refer to the new version of the manuscript.

Reviewer #1

- Major comment #1:
  - Thank you for adding Table 1. In the legend, can you be more specific about the two columns for "This paper" (I suppose the left one is for E3SM-Arctic-OSI and the right one for E3SM-LR-OSI) and about the two colums for Petersen et al. 2019 (I don't know what the two columns refer to).
  - Why isn't the RASM simulation described in this table?
  - Also in the text where you introduce Table 1, L112-113, please give a short written description of the differences so to help the reader understanding Table 1.

  Thank you for suggesting this. Indeed the original table was a bit unclear. The table has now been updated, reformatted, and described properly. We have also added the RASM simulation details to it.

- Major comment #2: Please provide some details about how parameterizations other than GM vary with resolution; if they don't please specify that.

  We have added the following sentence on line 103: "Other parameterizations used in MPAS-ocean are invariable with horizontal resolution.".

- Major comment #3:
  - Please add details in the text to justify the use of 3 cycles only.

    Done. Please see lines 122-125.

  - Furthermore, for me a "trend" is a variation of a quantity per unit of time. I think you should change on L141 "a cooling trend by up to 0.5°C" by "a cooling persistent anomaly of up to 0.5°C".

    Done.

  - Why is the upper ocean freshening more concerning than the other, especially more than the OHC 0-700 one?

    Indeed it is not more concerning than the other. We have now removed that sentence entirely and added instead: "Overall, the top-to-bottom trends are all reduced during the third cycle, whereas the upper ocean warming and freshening are both still present towards the end of the simulation." (lines 153-154).

- Major comment #5: ... I think L.175 "current GM implementation in MPAS-Ocean" is confusing. Maybe replace "current GM implementation in MPAS-Ocean" with "activation of GM in MPAS-Ocean".

  We can see how that sentence is confusing and have decided to remove it from the text.

- Major comment #7: Please add in the text some of your conclusions reported to the reviewer, even if you don't add Figures 1 and 2 from your reply.

  We have now added the following paragraph on lines 257-261: "The model climatologies are computed over the last 12 years of the third JRA cycle. A comparison with climatologies computed on an analogous period of the first cycle (not shown) indicate that, while some T and S changes are apparent below the Atlantic Water layer in Fram Strait and the BSO, and in the West Greenland Current in Davis Strait, the overall structure of the gateways stratification is quite consistent between the first and third JRA cycle, and consequently the velocity structures are also very comparable.".

- Major comment #8: ... Please add in the text some of your conclusions reported to the reviewer, i.e. the fact that the drift affects the stratification over the whole Arctic and locally over the Canada Basin only.

  We had actually already added this in the text, and then forgot to note it in the response to the reviewers.. Apologies for that. Please see lines 315-317.

- Major comment #9: Is the fact that salt/fresh water fluxes assume that sea ice has salinity of 4 psu mentioned somewhere in the text? If not do so, as the reviewer explicitly asked for that information.

  We added the following text on lines 104-106: "The version of MPAS-Seaice used in this paper and the way that the sea ice and ocean components are coupled together are fully described in Turner et al. (2022) and Petersen et al. (2019); we have not changed any default MPAS-Seaice parameter for the purposes of the present effort.".

- Major comment #11: Can you add something about this in the text?

  We are a bit concerned that adding something like this in the text may be seen as too speculative, especially now that we believe that the appropriate corrections were applied over sea-ice in the JRA55-do forcing (see next comment below).

- Major comment #12: I don't see anywhere in the text that you version v1.3 of the JRA55-do data set, can you add this and also something about the fact that you don't do any correction over ice?

  We had mentioned the JRA55-do version in the text (line 108). Regarding the correction issue: upon further investigation (Tsujino et al. 2018), we believe that the JRA55-do forcing used in the paper does indeed include corrections over sea-ice with respect to the JRA55-raw data set (see, for example, the comparison between shortwave and longwave radiation between JRA55-do and JRA55-raw and other data sets in Fig. 15 of Tsujino et al. 2018). We apologize for misstating that in our previous reply to Reviewer 1.

- Minor comments #2 and #3: Can you clarify these details somewhere in the text?

  Please see lines 104-106.

- Minor comment #12: Thanks for adding the profiles for the last 12 years of the first JRA cycle as dark red dashed lines in Figures 14, 15. In your reply, you write "They give a good idea of how the stratification changes from cycle to cycle over the whole Arctic and locally over the Canada Basin only."; can you explicitly detail your observations on this in the text?

  Please see lines 315-317.

- Comment #4: These are interesting remarks; please add something about this in the text.

  Good point. We have now added the following paragraph on lines 48-53: "A similar configuration to this, but with Arctic regional refinement of 6 km was also considered initially; that simulation, while being approximately three times more expensive than the one described in this paper, did not produce any significant improvements in the Arctic and subarctic ocean and sea-ice representation. We concluded that a resolution of at least 3 km is necessary to really resolve the local Rossby radius of deformation in most of the Arctic, and we plan to actively work on such very high resolution E3SM-Arctic configuration in the near future.".

My additional comments:

- Table 2: The captions mention E3SM-Arctic-OSI, E3SM-LR-OSI and observations but not RASM, which is indeed included in the Table; please modify the captions.

  Done.

- L 409: remove "they" at the end of the line.

  Done.

---

## Author Response (AR3)

**Response to editor's comment (Mar 15) for: An evaluation of the E3SMv1-Arctic Ocean/Sea Ice Regionally Refined Model**

March 15, 2022

Please note that the editor's comment is indicated in blue.

The only point I would like to raise is about the sentence you added: "The choice of three cycles is mostly the result of a trade off between reaching a trend reduction in global, top-to-bottom integrated fields such as temperature (T), salinity (S), ocean heat content (OHC), and sea-ice time series, and avoiding excessive drift of these same fields at the same time.". I don't understand this sentence as I don't understand the concept of a trade off between "reaching a trend reduction" and "avoiding excessive drift". Please clarify or simply remove this sentence and keep the one on the computational resources?

Thank you for the suggestion. We have now rephrased this sentence as follows (lines 122-125): "The choice of three cycles was mostly constrained by the availability of computational resources when these simulations were performed. We also compared trends of fields of interest during the second and third cycles, and, as the results shown later in the paper will elucidate, we were sufficiently satisfied that such trends remained mostly stable between the second and third cycle.", which we think is clearer and also better reflects our own response to the original Reviewer's comment.